# FINE-TUNING LANGUAGE MODELS WITH ADVANTAGE-INDUCED POLICY ALIGNMENT

## ABSTRACT

Reinforcement learning from human feedback (RLHF) has emerged as a reliable approach to aligning large language models (LLMs) to human preferences. Among the plethora of RLHF techniques, proximal policy optimization (PPO) is of the most widely used methods. Despite its popularity, however, PPO may suffer from instability and poor sample efficiency. We show that these issues can be alleviated by a novel algorithm that we refer to as Advantage-Induced Policy Alignment (APA), which leverages a squared error loss function based on the estimated advantages. We demonstrate empirically that APA consistently outperforms PPO in language tasks by a large margin, when a separate reward model is employed as the evaluator. In addition, compared with PPO, APA offers a more stable form of control over the deviation from the model's initial policy, ensuring that the model improves its performance without collapsing to deterministic output. In addition to empirical results, we also provide a theoretical justification supporting the design of our loss function.

## 1 INTRODUCTION

Reinforcement learning from human feedback (RLHF, or preference-based reinforcement learning) (Knox and Stone, 2008; Wirth et al., 2017) has delivered significant empirical successes in several fields, including games (Christiano et al., 2017), robotics (Sadigh et al., 2017; Kupcsik et al., 2018), recommendation systems (Maghakian et al., 2022). Recently, RLHF has also exhibited striking potential for integrating human knowledge with large language models (Ziegler et al., 2019; Ouyang et al., 2022; OpenAI, 2023; Beeching et al., 2023; Zhu et al., 2023; Bai et al., 2022b). To employ RLHF in the training pipeline of language models, a common protocol is as follows.

- **Pre-training (PT)**: training the language model on a large amount of unlabeled or weakly labeled text data to produce general features and patterns that can be useful for downstream tasks (Vaswani et al., 2017; Devlin et al., 2018; Brown et al., 2020);

- **Supervised fine-tuning (SFT)**: training the model on a smaller amount of curated data to improve the performance and accuracy of the model on specific tasks;

- **Reinforcement learning with human feedback (RLHF)**: using a human-labeled dataset together with reinforcement learning (RL) algorithms to further align the model with complex and subjective human values or preferences (Ziegler et al., 2019; Ouyang et al., 2022).

Both PT and SFT rely on the use of distributional loss functions, such as cross entropy, to minimize the distance between the text distributions in the training dataset and in the model output (Vaswani et al., 2017; Devlin et al., 2018; Brown et al., 2020). Such a simple strategy is not viable, however, for the RLHF stage. As the ultimate target is to make the language model output conform to human linguistic norms, which are difficult to define or quantify, researchers usually resort to a *reward model* that is trained separately from the language model on a meticulously collected, human-labeled dataset (Ouyang et al., 2022). Such a reward model produces a scalar score for each generated response, and is, therefore, able to provide the language model with online feedback. This accessibility to online feedback allows the language model to be trained via reinforcement learning (RL), giving rise to the RLHF stage.

Among the RL techniques that are applied to language models, one of the most prominent algorithms is proximal policy optimization (PPO) (Schulman et al., 2017). Despite the acclaimed effectiveness of PPO (Ouyang et al., 2022; Stiennon et al., 2020; Nakano et al., 2021), it suffers from instability and poor sample efficiency. The family of policy gradient algorithms suffers from slow convergence and can yield poor ultimate policies (Yuan et al., 2022; Dong et al., 2022).

To address such issues, we introduce a novel algorithm, *Advantage-Induced Policy Alignment* (APA), which leverages a squared error loss function that directly aligns the output policy of the language model with a target policy in each training epoch. The target policy combines the initial language model policy before the RLHF stage and a correction term based on the advantage function estimated from online samples.

We compare APA with PPO and advantage weighted regression (AWR) both theoretically and empirically. At a high level, the two existing algorithms (PPO and AWR) solve a KL-constrained policy optimization problem, relying on the estimated importance ratio between consecutive policies to compute the policy gradient. On the other hand, APA uses squared error to regularize the deviation of model policy and no longer requires estimating the importance ratio. We show that such differences bring huge benefits in terms of sample efficiency.

To demonstrate the efficacy of APA empirically, we apply APA, PPO and AWR to fine-tuning up to 7B language models, using the human-labeled Anthropic Helpfulness and Harmlessness dataset (Ganguli et al., 2022) and the StackExchange Beeching et al. (2023) dataset. We first evaluate using the reward model, trained on the same dataset to produce a scalar reward for each prompt-response pair. We also evaluate the human preferences of the resulting language model using GPT-4 to demonstrate the effectiveness of the algorithm. Our empirical results highlight three major advantages of APA over PPO:

(i) **APA is more sample-efficient**. Fine-tuned on the same number of samples, the language model obtained via APA scores consistently higher on the evaluation set than the one obtained with PPO.

(ii) **APA affords steadier control over the deviation from the language model's initial policy**. Measured by KL divergence, the deviation of the ultimate policy generated by APA is comparable with that of PPO, yet APA is less prone to sudden performance degradation during training, which is occasionally observed in PPO. Note that previous study has shown that the control over deviations from the initial policy is critical in preventing over-optimization on reward models (Gao et al., 2022).

(iii) **APA has fewer hyperparameters**. The loss function in APA involves only one major tunable parameter for KL control, whereas in PPO one has to carefully calibrate the combination of various extra hyperparameters, such as the clipping ranges for importance ratio and value estimates, and the coefficients of the KL controller.

More broadly, this work is related to the line of literature on leveraging ideas from RL to improve the performance of language models. A few notable examples in this literature include Paulus et al. (2017), who propose a loss function based on the policy gradient objective to tackle the abstractive summarization task, using ROUGE scores as reward; and Snell et al. (2022), who present the implicit language $Q$-learning (ILQL) algorithm to facilitate learning from offline human-labeled samples without a reward model. A thorough comparison between different RL algorithms is also made in Ramamurthy et al. (2022) on GRUE benchmarks. There have been some alternative frameworks of RLHF that replaces PPO with SFT on best generated sample (Yuan et al., 2023), or a direct preference-based offline learning (Rafailov et al., 2023).

The remainder of this paper is organized as follows. In Section 2, we introduce our notation. In Section 3, we formally specify the algorithm APA, and discuss the intuitions behind the algorithmic elements. Experimental results are presented in Section 4. Section 5 concludes by summarizing and discussing the experimental results.

## 2 PRELIMINARIES

In this section, we overview the standard RL setting in Section 2.1, and discuss how language model training fits into this setting in Section 2.2. We use the following notation. For a positive integer $n$,

we will use the bracket notation $[n]$ to refer to the set of integers $\{1, \ldots, n\}$; for a finite set $\mathcal{Z}$, we denote by $\Delta(\mathcal{Z})$ the set of probability distributions on $\mathcal{Z}$, and $|\mathcal{Z}|$ the cardinality of $\mathcal{Z}$. We use $\mathbb{B}^d$ to denote the unit ball in $d$-dimensional space.

## 2.1 REINFORCEMENT LEARNING

Reinforcement learning (RL) captures the interaction between an agent and an environment via the formalism of a Markov decision process (MDP). We consider a finite-horizon MDP represented by a tuple $M = (\mathcal{S}, \mathcal{A}, H, P, r, \rho)$, where $\mathcal{S}$ is a finite state space, $\mathcal{A}$ is a finite action space, $H$ is the horizon, $P : \mathcal{S} \times \mathcal{A} \mapsto \Delta(\mathcal{S})$ is a probability transition matrix, $r : \mathcal{S} \times \mathcal{A} \mapsto [0, 1]$ is a reward function, and $\rho : \mathcal{S} \mapsto \Delta(\mathcal{S})$ is the initial state distribution. When the agent takes action $a$ in state $s$ at step $h$, it receives a scalar reward $r(s, a)$, and transitions to a state $s'$, where $s'$ is drawn from distribution $P(\cdot|s, a)$. Each episode consists of $H$ consecutive steps. At the end of an episode, the agent is reset to a state drawn from $\rho(\cdot)$, and a new episode begins.

A policy $\pi : \mathcal{S} \mapsto \Delta(\mathcal{A})$ is a function that maps a state to a distribution over actions. The value function $V^\pi : \mathcal{S} \mapsto \mathbb{R}$ of policy $\pi$ is defined as the expected sum of discounted rewards when the agent starts from initial state $s$ and follows policy $\pi$ throughout the episode. Let $\gamma \in [0, 1]$ be the discount factor. For any $s \in \mathcal{S}$, we have

$$V^\pi(s) := \mathbb{E}\left[\sum_{\tau=0}^{H} \gamma^\tau r(s_\tau, a_\tau) \mid s_0 = s, a_\tau \sim \pi(\cdot|s_\tau), s_{\tau+1} \sim P(\cdot \mid s_\tau, a_\tau)\right].$$

Given a policy $\pi$, the state-action value function, also known as the $Q$-function, can be defined analogously. For state $s \in \mathcal{S}$ and $a \in \mathcal{A}$, we have

$$Q^\pi(s, a) := \mathbb{E}\left[\sum_{\tau=0}^{H} \gamma^\tau r(s_\tau, a_\tau) \mid s_0 = s, a_0 = a, a_\tau \sim \pi(\cdot|s_\tau), s_{\tau+1} \sim P(\cdot \mid s_\tau, a_\tau)\right].$$

We also define the important notion of an *advantage function*. For a policy $\pi$, state $s$ and action $a$, the advantage, defined as

$$\mathsf{Adv}^\pi(s, a) = Q^\pi(s, a) - V^\pi(s),$$

quantifies the extra value that is obtained by replacing the immediate action prescribed by $\pi$ with the action $a$, when the agent is in state $s$ at step $h$.

We also define the *occupancy measures* $d^\pi_{\text{state}} : \mathcal{S} \mapsto [0, 1]$ and $d^\pi_{\text{action}} : \mathcal{S} \times \mathcal{A} \mapsto [0, 1]$ as

$$d^\pi_{\text{state}}(s) := \frac{1}{H}\sum_{h=1}^{H}\mathbb{P}(s_h = s \mid \pi) \quad \text{and} \quad d^\pi_{\text{action}}(s, a) := \frac{1}{H}\sum_{h=1}^{H}\mathbb{P}(s_h = s, a_h = a \mid \pi),$$

where $\mathbb{P}(\cdot \mid \pi)$ signifies that all actions are drawn from $\pi$. To avoid clutter, we overload the notation $d^\pi$ such that $d^\pi(s)$ refers to $d^\pi_{\text{state}}(s)$, and $d^\pi(s, a)$ refers to $d^\pi_{\text{action}}(s, a)$.

## 2.2 LANGUAGE MODEL AS REINFORCEMENT LEARNING AGENT

In its simplest form, a language model receives as input a sequence of tokens $(x_1, \ldots, x_n)$, and generates a distribution over the next token $x_{n+1}$. All tokens lie in a finite set $\mathcal{X}$. Whenever the agent selects a token that represents the completion of a response (e.g., the end-of-sequence token), or the total number of tokens reaches a specific limit, the entire sequence is scored by a reward model, which produces a scalar reward $r$.

Comparing with the RL formulation in Section 2.1, a language model can be viewed as an agent that operates in an environment with state space $\mathcal{S} = \bigcup_{k=0}^{H} \mathcal{X}^k$ and action space $\mathcal{A} = \mathcal{X}$, where $H$ is the maximum number of tokens. The transitions are always deterministic, with the next state equal to the concatenation of all the previous tokens and the current token $P(s_{h+1} = (x_1, \cdots, x_k) \mid s_h = (x_1, \cdots, x_{k-1}), a_h = x_k) = 1$. Traditionally, each episode involves the generation of one complete sequence, and a reward is delivered only when an episode terminates. In this context, fine-tuning is equivalent to improving the agent policy $\pi$. The field of RL offers a formidable arsenal for this task. In this work, we will focus on policy-based RL algorithms, which parameterize the set of agent

policies by a set of parameters $\theta$ and optimize in the parameter space. In what follows, we will omit the step index $h$, as its information is already encoded in each state.

We note that most transformer-based language models map a state (context) $s$ and an action (next token) $a$ to a logit $q_\theta(s, a)$, and the next token is sampled according to the distribution induced by the logits $\{q_\theta(s, a)\}_{a \in \mathcal{A}}$. This naturally gives rise to the following parameterization of language model policy:

$$\pi_\theta(a \mid s) = \frac{\exp(q_\theta(s, a))}{\sum_{a \in \mathcal{A}} \exp(q_\theta(s, a))}.$$

## 3 FINE-TUNING BASED ON REINFORCEMENT LEARNING

As is mentioned in Section 1, the RLHF stage is usually composed of two steps. First, a reward model is trained from a human-labeled dataset. An RL algorithm is then applied to improve the language model policy, using the rewards generated by the reward model. Here we focus mainly on the second step with a given reward function.

We summarize a typical policy-based RL algorithm in Algorithm 1. In practice, the parameter update in Equation (1) usually involves several gradient steps rather than a full minimization.

---

**Algorithm 1** Policy Gradient

1: **Input:** An initial policy parameter $\theta_0$, a given loss function $\mathcal{L}(\theta; \mathcal{D})$.
2: Set $\pi_0 = \pi_{\text{init}}$.
3: **For** iteration $t = 1, 2 \cdots, T$
4:     Roll out $\pi_{\theta_{t-1}}$ to produce dataset $\mathcal{D}_t = \left\{ (s_1^{(t)}, a_1^{(t)}, r_1^{(t)}), \cdots, (s_n^{(t)}, a_n^{(t)}, r_n^{(t)}) \right\}$
5:     Update policy parameter according to

$$\theta_t = \arg\min_\theta \mathcal{L}(\theta; \mathcal{D}_t). \tag{1}$$

---

In the remainder of this section, we discuss several potential choices for $\mathcal{L}(\theta; \mathcal{D})$, each targeting the goal of maximizing regularized advantages. We also introduce the new algorithm APA, and discuss the intuitions behind it.

As a first step, for each fixed state $s$, we consider the following KL-regularized optimization problem as a target of policy improvement:

$$\underset{\theta}{\text{maximize}} \, \mathcal{F}(\theta; s, \pi) := \mathbb{E}_{a \sim \pi_\theta(\cdot | s)}[\text{Adv}^\pi(s, a)] - \lambda \cdot \text{KL}\Big(\pi_\theta(\cdot \mid s) \big\| \pi_{\text{init}}(\cdot \mid s)\Big). \tag{2}$$

Here $\pi_{\text{init}}$ refers to the initial policy of the language model before the RLHF stage, $\pi$ is an arbitrary policy that we hope to improve upon. The first term in the objective function $\mathcal{F}(\theta; s, \pi)$ is an expected advantage, and to maximize the expected advantage, the agent is encouraged to move toward the optimal action in state $s$. The second term in $\mathcal{F}(\theta; s, \pi)$, a KL regularizer, controls the deviation of $\pi_\theta$ from $\pi_{\text{init}}$. Such regularization is essential, as language models are prone to over-optimization when rewards are generated by an imperfect reward model, a phenomenon observed in Gao et al. (2022). Combined, the single-state optimization problem in (2) aims at improving upon policy $\pi$ in state $s$ within the proximity of $\pi_{\text{init}}$.

The optimization (2) is usually broken down into multiple iterations. In each iteration, we maximize $\mathcal{F}(\theta; s, \pi_{\text{old}})$, where $\pi_{\text{old}}$ is the policy that the agent arrives at in the previous iteration. This technique, referred to as *Conservative Policy Iteration (CPI)*, was first presented in Kakade and Langford (2002). The optimization was subsequently generalized to KL-constrained and regularized methods referred to as *Trust Region Policy Optimization (TRPO)* (Schulman et al., 2015a) and Proximal Policy Optimization (PPO) (Schulman et al., 2017), respectively. In addition to these core methods, there have been several other policy optimization methods inspired by (2), with one notable example being the *Advantage-Weighted Regression (AWR)* method (Peng et al., 2019; Nair et al., 2020).

In the following subsection, we will discuss how $\mathcal{F}(\theta; s, \pi)$ is connected with the loss function $\mathcal{L}(\theta; \mathcal{D})$ in various algorithms, and propose a new proximal optimization problem whose solution

approximates that of (2). The loss function in APA will be based on this new proximal optimization problem.

## 3.1 PROXIMAL POLICY OPTIMIZATION

PPO leverages importance sampling to circumvent sampling from $\pi_\theta$, arriving at

$$\mathbb{E}_{a \sim \pi_\theta(\cdot|s)} \left[ \mathsf{Adv}^{\pi_{\mathsf{old}}}(s, a) \right] = \mathbb{E}_{a \sim \pi_{\mathsf{old}}(\cdot|s)} \left[ \frac{\pi_\theta(a \mid s)}{\pi_{\mathsf{old}}(a \mid s)} \mathsf{Adv}^{\pi_{\mathsf{old}}}(s, a) \right],$$

where the expectation on the right-hand side can be estimated in an unbiased manner from finite samples.

PPO also involves the following innovation: Instead of penalizing the expected advantage with the estimated KL-divergence as in (2), PPO directly subtracts the KL penalty term from the reward received by the agent. And one may also adaptively adjust the penalty weight $\lambda$ based on the deviation of $\pi_\theta$ from $\pi_{\mathsf{init}}$ (Schulman et al., 2017; Dhariwal et al., 2017; Ziegler et al., 2019). The KL-penalized reward is then used to estimate a new advantage function $\widehat{\mathsf{Adv}}$. To avoid ill-conditioned gradients caused by large values or importance ratio estimates, PPO applies clipping to the objective function. The final loss function is thus

$$\mathcal{L}^{\mathsf{PPO}}(\theta; \mathcal{D}) = -\frac{1}{|\mathcal{D}|} \sum_{(s,a) \in \mathcal{D}} \min \left\{ \frac{\pi_\theta(a \mid s)}{\pi_{\mathsf{old}}(a \mid s)} \widehat{\mathsf{Adv}}(s, a), \mathsf{clip} \left( \frac{\pi_\theta(a \mid s)}{\pi_{\mathsf{old}}(a \mid s)}, 1 - \epsilon, 1 + \epsilon \right) \widehat{\mathsf{Adv}}(s, a) \right\}.$$

Note that the loss function relies on extra tunable hyperparameters. The clipping also makes the estimator biased.

## 3.2 ADVANTAGE WEIGHTED REGRESSION

If the parameterized policy space $\{\pi_\theta\}$ contained all possible policies including the ground truth policy, the maximizer of $\mathcal{F}(\theta; s, \pi_{\mathsf{old}})$ (2) would induce a policy $\pi^\star$ that satisfies

$$\pi^\star(a \mid s) = \frac{1}{Z(s)} \pi_{\mathsf{init}}(a \mid s) \cdot \exp(\mathsf{Adv}^{\pi_{\mathsf{old}}}(s, a)/\lambda), \tag{3}$$

where $Z(s) = \sum_{a' \in \mathcal{A}} \pi_{\mathsf{init}}(a' \mid s) \cdot \exp(\mathsf{Adv}^\pi(s, a')/\lambda)$ is a normalizing factor. In the case that $\{\pi_\theta\}$ does not contain all policies, a natural way to maximize $\mathcal{F}(\theta; s, \pi_{\mathsf{old}})$ is to project $\pi^\star$ to $\{\pi_\theta\}$ with respect to KL-divergence. From (3),

$$\mathsf{KL}\big(\pi^\star(a \mid s) \| \pi_\theta(a \mid s)\big) = -\frac{\pi_{\mathsf{init}}(a \mid s)}{Z(s)} \exp \left( \frac{\mathsf{Adv}^{\pi_{\mathsf{old}}}(s, a)}{\lambda} \right) \log \big(\pi_\theta(a \mid s)\big) + C(s), \tag{4}$$

where $C(s)$ is a constant that does not depend on $\theta$.

To facilitate online update, AWR makes three changes from Equation (4):

- AWR replaces the first-round policy $\pi_{\mathsf{init}}$ with the previous-round policy $\pi_{\mathsf{old}}$. This ensures that one can utilize the new roll-out samples from previous-round policy to approximate (4).
- The KL-divergence in (4) only accounts for one state $s$. AWR minimizes a distribution of states $d^{\pi_{\mathsf{old}}}$.
- AWR approximates $Z(s) \approx 1$. We also provide a related discussion in Appendix A on why such an approximation is warranted.

These changes lead to the loss function introduced in AWR:

$$\mathcal{L}^{\mathsf{AWR}}(\theta) = -\mathbb{E}_{(s,a) \sim d^{\pi_{\mathsf{old}}}} \left[ \exp \big( \mathsf{Adv}^{\pi_{\mathsf{old}}}(s, a)/\lambda \big) \log \big( \pi_\theta(a \mid s) \big) \right]. \tag{5}$$

Given a finite dataset $\mathcal{D} = \{(s_i, a_i) : i = 1, \dots, n\}$ sampled from $d^{\pi_{\mathsf{old}}}$, the corresponding empirical loss can be written as

$$\mathcal{L}^{\mathsf{AWR}}(\theta; \mathcal{D}) = -\frac{1}{|\mathcal{D}|} \sum_{(s,a) \in \mathcal{D}} \exp \big( \mathsf{Adv}^{\pi_{\mathsf{old}}}(s, a)/\lambda \big) \log \big( \pi_\theta(a \mid s) \big). \tag{6}$$

For the well-specified case where the parameterized family $\{\pi_\theta\}$ contains the minimizing policy, the minimizer of the population loss is as follows:

$$\pi'(a \mid s) = \frac{\pi_{\text{old}}(a \mid s) \exp(\text{Adv}^{\pi_{\text{old}}}(s, a)/\lambda)}{\sum_a \pi_{\text{old}}(a \mid s) \exp(\text{Adv}^{\pi_{\text{old}}}(s, a)/\lambda)}. \tag{7}$$

Due to the discrepancies between the original target in Equation (4) and the final loss in Equation (5), one can see that the policy AWR converges to is different from the the original target in Equation (3). Furthermore, since $\pi_{\text{old}}$ changes in each round, the policy it converges to continues to change.

As we observe in Section 4 and Appendix E.3, AWR can be unstable in the online case due to this reason. This motivates us to introduce APA, which alleviates this issue, provably converges to the right target in Equation (3), and demonstrates great empirical performance.

### 3.3 Advantage-Induced Policy Alignment

To project the optimal policy $\pi^\star$ in (3) onto the parameterized policy space, we consider another distance instead of KL-divergence. In APA, we employ the squared error between log probabilities in place of the KL-divergence:

$$\big(\log \pi^\star(a \mid s) - \log \pi_\theta(a \mid s)\big)^2 = \Big(\log \pi_\theta(a \mid s) + \log Z(s) - \text{Adv}^{\pi_{\text{old}}}(s, a)/\lambda - \log \pi_{\text{init}}(a \mid s)\Big)^2.$$

Similar to the approximation in AWR, we also apply $Z(s) \approx 1$, and minimize the expected loss under a state distribution $d^{\pi_{\text{old}}}$ in each round, giving rise to the following population loss:

$$\mathcal{L}^{\text{APA}}(\theta) = \mathbb{E}_{(s,a) \sim d^{\pi_{\text{old}}}} \Big[\Big(\log \pi_\theta(a \mid s) - \text{Adv}^{\pi_{\text{old}}}(s, a)/\lambda - \log \pi_{\text{init}}(a \mid s)\Big)^2\Big]. \tag{8}$$

The empirical loss on a finite dataset $\mathcal{D}$ sampled from $d^{\pi_{\text{old}}}$ is thus

$$\mathcal{L}^{\text{APA}}(\theta; \mathcal{D}) = \frac{1}{|\mathcal{D}|} \sum_{(s,a) \in \mathcal{D}} \Big(\log \pi_\theta(a \mid s) - \text{Adv}^{\pi_{\text{old}}}(s, a)/\lambda - \log \pi_{\text{init}}(a \mid s)\Big)^2. \tag{9}$$

Assuming that the parameter space is $\Theta = \mathbb{B}^d$ and that the parameterized policy space is well-specified such that $\pi^\star \in \{\pi_\theta \mid \theta \in \Theta\}$, where $\pi^\star$ is defined in Equation (3), we can establish theoretically that the empirical loss is a reasonable surrogate for the population loss.

**Theorem 1.** *Let $\theta^\star \in \arg\min_{\theta \in \Theta} \mathcal{L}^{\text{APA}}(\theta)$ be a minimizer of the population loss. Then*

$$\pi_{\theta^\star}(a \mid s) = \pi^\star(a \mid s), \quad \forall(s, a) \in \text{supp}(\pi_{\text{old}}).$$

*Furthermore, let $\hat{\theta} \in \arg\min_{\theta \in \Theta} \mathcal{L}^{\text{APA}}(\theta, \mathcal{D})$ be an empirical loss minimizer. Assume that $\min(\pi_\theta(a \mid s), \pi_{\text{init}}(a \mid s)) \geq B_1$ and $|\text{Adv}(s, a)| \leq B_2$ for any $s, a$, and that $\log(\pi_\theta)$ is L-Lipschitz with respect to $\theta$ under $\ell_2$-norm for any $s, a$. Then for all $\delta > 0$, with probability at least $1 - \delta$, for some universal constant $C$,*

$$\mathcal{L}^{\text{APA}}(\hat{\theta}) \leq CL(B_2 - \log(B_1))^2 \sqrt{\frac{d \log(nL/\delta)}{n}}.$$

The proof is deferred to Appendix F. From the theorem, we see that the minimizer of the population APA loss is exactly the target policy $\pi^\star$ if the policy $\pi_{\text{old}}$ is supported on all state-action pairs. In contrast, as we discussed earlier, convergence properties of the PPO and AWR algorithms have not yet been established.

We also provide alternative interpretations of the proposed loss in terms of $f$-divergence in Appendix B and alternative loss inspired by soft-Q learning in Appendix and C.

## 4 Experimental Results

In our implementation of all of the algorithms that we test, including APA, we define the advantage function to be $\text{Adv}^{\pi_{\text{old}}}(s, a)$, which is estimated from data. We use the

same generalized advantage estimation approach to estimate the advantage as discussed in earlier work Mnih et al. (2016); Schulman et al. (2015b). In particular, for the rollout $(s_0, a_0, r_0, s_1, a_1, r_1, \cdots, s_{T-1}, a_{T-1}, r_{T-1}, s_T)$, the generalized advantage estimator is

$$\hat{A}^{\pi_{\text{old}}}(s_t, a_t) = \delta_t + \lambda\gamma\delta_{t+1} + \cdots + (\lambda\gamma)^{T-1}\delta_{T-1},$$
$$\text{where } \delta_t = r(s_t, a_t) + \gamma V^{\pi_{\text{old}}}(s_{t+1}) - V^{\pi_{\text{old}}}(s_t).$$

Here the value function is another standalone network that we fit throughout the training process with a squared loss, $\hat{\mathcal{L}}_V(\mathcal{D}) = \sum_{s_i, a_i}(V(s_i) - \hat{A}^{\pi_{\text{old}}}(s_i, a_i) - V^{\pi_{\text{old}}}(s_i))^2$. Thus the overall loss function is

$$\mathcal{L}_\theta^{\text{APA}}(\mathcal{D}) = \hat{\mathcal{L}}^{\text{APA}}(\mathcal{D}) + \eta \cdot \hat{\mathcal{L}}_V(\mathcal{D})$$
$$\mathcal{L}_\theta^{\text{AWR}}(\mathcal{D}) = \hat{\mathcal{L}}^{\text{AWR}}(\mathcal{D}) + \eta \cdot \hat{\mathcal{L}}_V(\mathcal{D}).$$

For the implementation of PPO, we use the PPO2 version from Dhariwal et al. (2017), with the adaptive KL controller from Ziegler et al. (2019). We implement PPO with the same hyperparameters as the implementation in `trlX`[1], which also follows default hyperparameters suggested by Schulman et al. (2017). The main difference between our version of PPO and that in `trlX` is that we create a completely separate value network rather than creating a value head on top of the language model. In APA, we take $\lambda = 0.1$ to impose a weaker constraint on the KL coefficient. For AWR, we find that setting $\lambda = 0.1$ leads to an explosion of the loss; thus we take $\lambda = 1$ to stabilize training.

## 4.1 RESULTS ON THE STACKEXCHANGE DATASET

In this section, we present our experimental results with StackExchange dataset. This dataset includes questions and their corresponding answers from the StackExchange platform (including StackOverflow for code and many other topics). The answers are then voted by the users on the platform and an accepted answer is labeled. Following Beeching et al. (2023); Askell et al. (2021), we assign a score to each answer depending on the number of upvotes:

$$\text{score} = \begin{cases} -1, & \text{upvotes} \leq 0, \\ 1 + \lfloor\log_2(1 + \text{upvotes}) + 0.5\rfloor, & \text{if the questioner accepted the answer,} \\ \lfloor\log_2(1 + \text{upvotes}) + 0.5\rfloor, & \text{otherwise.} \end{cases}$$

We used the pre-processed dataset provided in Beeching et al. (2023) for all SFT, reward modeling and RL training purposes, available in the HuggingFace Datasets as `lvwerra/stack-exchange-paired`[2]. We use LLaMA-7B Touvron et al. (2023) models for this experiment. We use Low-Rank Adaptation (LoRA) method Hu et al. (2021) to reduce the memory consumption while training. We used 8xA100 GPUs for our experiments. The hyper-parameters for this experiment are listed in the Appendix D.4. Fig. 2 shows the reward on the left and KL divergence from the initial policy for the three algorithms, PPO, APA and AWR. We adjust the hyper-parameters to achieve similar KL divergence values, allowing us to compare the rewards for various algorithms. In the case of AWR, each hyper-parameter set displayed some level of instability. Clearly, APA quickly converges to a higher reward than PPO and AWR.

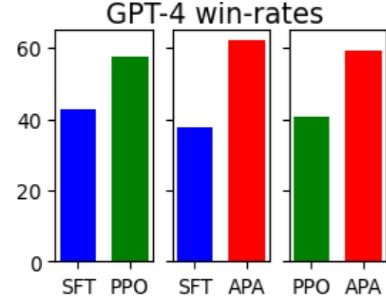

Figure 1: Win rates computed by GPT-4 for StackExchange dataset for models trained by SFT, PPO and APA. Compared to SFT and PPO, APA trained models generated better responses.

**GPT-4 Evaluation:** We conduct a GPT-4 evaluation to evaluate the models trained by different RL methods on StackExchange dataset. GPT-4 compares the outputs produced by two models, using a reference (chosen) response as a basis for comparison. Fig. 1 shows the win-rates for comparing SFT vs PPO, SFT vs APA and PPO vs APA models. APA consistently outperforms the other two models.

[1] https://github.com/CarperAI/trlx
[2] https://huggingface.co/datasets/lvwerra/stack-exchange-paired

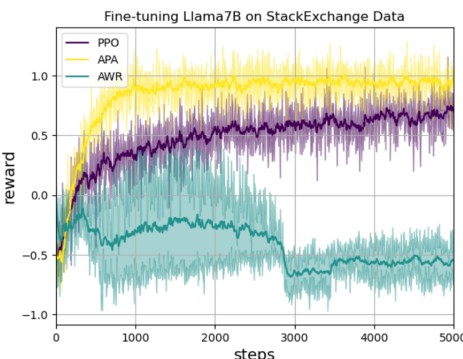 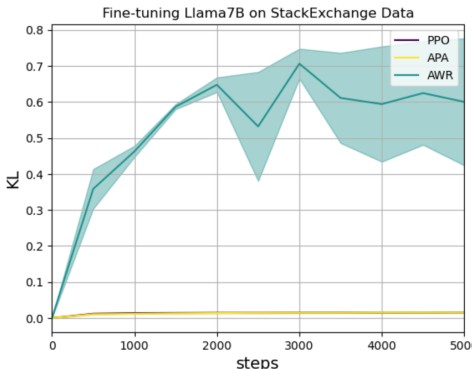

Figure 2: Comparison of the performance of three methods on the StackExchange dataset. Left: The $x$-axis represents the total steps, which are proportional to the amount of data used in the training procedure. The $y$-axis is the reward computed by the same reward model during training. Right: The $x$-axis represents the total steps. The $y$-axis is the KL divergence between the trained model and the initial model.

## 4.2 RESULTS ON THE HH DATASET

In this section, we compare PPO, AWR and APA on the human-labeled Helpfulness and Harmlessnes (HH) dataset from Bai et al. (2022a).[3] We fine tune three models, including `Dahoas/pythia-125M-static-sft`,[4] `Dahoas/pythia-1B-static-sft`,[5] and `Dahoas/pythia-6B-static-sft`.[6] All the models have gone through supervised fine-tuning with labeled prompt-response pairs, similar to the protocol in Ouyang et al. (2022) and Ramamurthy et al. (2022). We present the performance of the RL algorithms for `pythia-125M` and `pythia-1B` in Fig. 3. It shows that after some steps, PPO's performance begins to deteriorate while APA and AWR are stable. APA achieves the higher reward while maintaining KL divergence from the initial policy smaller. The details about hyper-parameters used for training and additional results for larger models are discussed in Appendix D.1.

## 5 CONCLUSIONS

In this paper, we study the problem of online policy optimization in RLHF. We benchmark the performance of existing algorithms PPO and AWR, and introduce a new method, APA, which has a theoretical convergence guarantee and affords several advantages over existing algorithms. The key takeaways from our study can be summarized as follows.

**Stability.** As we discussed in Section 1, one of the challenges in RLHF is instability. It is crucial in RL algorithms to impose control on the divergence between new policies and the initial policy after SFT. However, the clipping of the objective function and the adaptive KL controller can make the behavior of PPO unstable; for AWR, the update in (7), which reweighs the previous policy by a multiplicative factor in each iteration, also has unknown ramifications. APA, on the other hand, provably converges to $\pi^\star$ when the advantage function is fixed, which is close to the initial policy in KL divergence. From the experimental results, we see that APA is able to provide better and easy-to-adjust KL control by explicitly tuning the hyperparameter $\lambda$. Our experiments reveal different levels of instability for PPO and AWR. Specifically, PPO suffers from occasional performance degradation whenever the model policy diverges too much from the initial policy $\pi_{\text{init}}$, and such effect is more pronounced for smaller models. We attribute this to the KL controller in PPO. In Appendix E, we demonstrate that PPO can achieve a similar sample efficiency as APA without the KL penalty, albeit at the cost of weaker KL efficiency.

---

[3] `https://huggingface.co/datasets/Dahoas/static-hh`
[4] `https://huggingface.co/Dahoas/pythia-125M-static-sft`
[5] `https://huggingface.co/Dahoas/pythia-1B-static-sft`
[6] `https://huggingface.co/Dahoas/pythia-6B-static-sft`

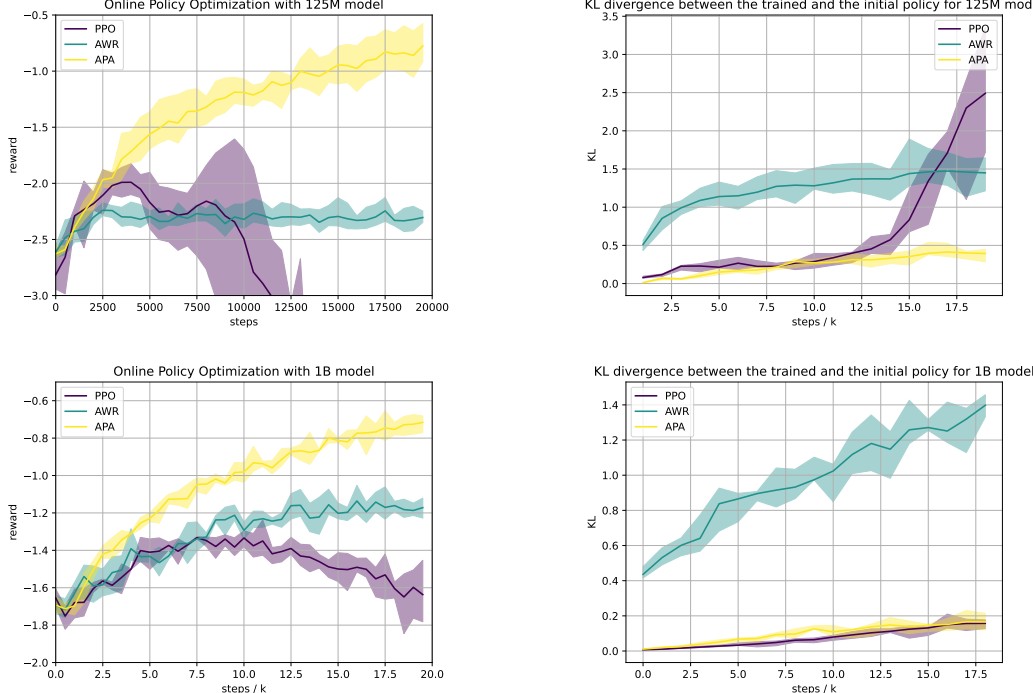

Figure 3: Comparison of the performance of three methods on the HH dataset. Left: The $x$-axis represents the total steps, which are proportional to the amount of data used in the training procedure. The $y$-axis is the reward evaluated by the same reward model. Right: The $x$-axis represents the total steps. The $y$-axis is the KL divergence between the trained model and the initial model.

**Sample efficiency.** With the same level of control over KL-divergence, APA shows higher sample efficiency than PPO and AWR. One possible explanation is that in both PPO and AWR, policy improvement critically depends on using finite samples to reconstruct the sampling policy $\pi_{\mathsf{old}}$, whereas in APA, minimizing the population loss (8) hinges less on the reconstruction of $\pi_{\mathsf{old}}$. In fact, the APA population loss (8) can be effectively minimized as long as the dataset $\mathcal{D}$ has a decent coverage over state-action pairs that are frequently visited by $\pi_{\mathsf{old}}$. For more discussions on sample efficiency, please refer to Appendix B.

**Online vs. offline learning.** Our experiments primarily examine the online case, where new data can be generated during the training process. The offline setting, where a fixed dataset is given and new samples are not available, may yield qualitatively different results. In particular, suppose that the offline dataset consists of rollouts from a policy $\pi_{\mathsf{off}}$. In this case, if it were trained with infinitely many samples, AWR would converge to the policy specified in (3). However, the performance of APA may suffer from distribution shift because it can only learn from state-action pairs covered by $\pi_{\mathsf{off}}$, and there is no guarantee that the learned policy performs well on the state-action pairs visited by the current policy. Such distribution mismatch can lead to a significant performance drop for APA, as we observe in Appendix E.3. We also observe that AWR typically outperforms ILQL for offline learning, although both perform poorly with larger models.

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

## A  ARGUMENT FOR $Z(s) \approx 1$

Note that in both advantage-weighted regression and advantage-based squared loss, we approximate $Z(s)$ with 1. Here we justify why this does not hurt the performance.

Consider an infinitesimal scenario where $|\mathsf{Adv}/\lambda| \ll |\log \pi_{\mathsf{init}}|$. In the scenario of language model, this is usually true since $\pi_{\mathsf{init}}$ is supported on approximately $50k$ distinct tokens and can be very close to zero, while $\mathsf{Adv}/\lambda$ can be adjusted to small numbers by adjusting $\lambda$.

In this case, we have

$$
\begin{aligned}
Z(s) &= \sum_{a \in \mathcal{A}} \pi_{\mathsf{init}}(a \mid s) \exp(\mathsf{Adv}(s,a)/\lambda) \\
&= \mathbb{E}_{a \sim \pi_{\mathsf{init}}}[\exp(\mathsf{Adv}(s,a)/\lambda)] \\
&= \mathbb{E}_{a \sim \pi_{\mathsf{init}}}[1 + \mathsf{Adv}(s,a)/\lambda + o(\mathsf{Adv}^2(s,a)/\lambda^2)].
\end{aligned}
$$

This advantage is usually estimated as $\mathsf{Adv}^{\pi_{\mathsf{old}}}$, which can be close to $\mathsf{Adv}^{\pi_{\mathsf{init}}}$. And we have

$$
\mathbb{E}_{a \sim \pi_{\mathsf{init}}}[\mathsf{Adv}^{\pi_{\mathsf{old}}}(s,a)/\lambda] \approx \mathbb{E}_{a \sim \pi_{\mathsf{init}}}[\mathsf{Adv}^{\pi_{\mathsf{init}}}(s,a)/\lambda] = 0.
$$

Thus we know that

$$
Z(s) \approx 1 + \mathbb{E}_{a \sim \pi_{\mathsf{init}}}[o(\mathsf{Adv}^2(s,a)/\lambda^2)] \approx 1.
$$

In practice, we observe that the squared loss decreases very slowly due to a small learning rate ($8e{-}6$). This suggests that the policy changes very slowly, which is another reason why the normalizing factor is not important.

## B  ALTERNATIVE INTERPRETATION OF APA

Recall that APA can be written as

$$
\mathcal{L}^{\mathsf{APA}}(\theta) = \mathbb{E}_{(s,a) \sim d^{\pi_{\mathsf{old}}}}\left[\left(\log \pi_\theta(a \mid s) - \log \pi^\star(a \mid s)\right)^2\right],
$$

where $\pi^\star = \pi_{\mathsf{init}} \cdot \exp(\mathsf{Adv}/\lambda)$. In the case when $\pi_{\mathsf{old}}$ is close to $\pi_\theta$, minimizing the squared loss in APA is equivalent to minimizing the following distance between $\pi^\star$ and $\pi_\theta$:

$$
d(\pi^\star(\cdot \mid s), \pi_\theta(\cdot \mid s)) = \sum_a \pi_\theta(a \mid s) \left(\log\left(\frac{\pi_\theta(a \mid s)}{\pi^\star(a \mid s)}\right)\right)^2.
$$

This can be viewed as a new $f$-divergence with $f(x) = x \log^2(x)$. We can show by Cauchy-Schwarz that this is always a upper bound for the KL divergence:

$$
\begin{aligned}
d(\pi^\star(\cdot \mid s), \pi_\theta(\cdot \mid s)) &= \left(\sum_a \pi_\theta(a \mid s)\right)\left(\sum_a \pi_\theta(a \mid s)\left(\log\left(\frac{\pi_\theta(a \mid s)}{\pi^\star(a \mid s)}\right)\right)^2\right) \\
&\geq \sum_a \pi_\theta(a \mid s)\left|\log\left(\frac{\pi_\theta(a \mid s)}{\pi^\star(a \mid s)}\right)\right| \\
&\geq \sum_a \pi_\theta(a \mid s)\log\left(\frac{\pi_\theta(a \mid s)}{\pi^\star(a \mid s)}\right).
\end{aligned}
$$

## C  AN ALTERNATIVE LOSS WITH SOFT-Q LEARNING

In this section, we discuss a different fine-tuning algorithm that is inspired by the soft Q-learning perspective (Haarnoja et al., 2018; Guo et al., 2021). However, we do not see a strong empirical performance of the proposed algorithm compared with APA. We include the theoretical analysis and intuition here for potential future trials.

## C.1 PRE-TRAINING

We view the pre-training (and the SFT) procedure as max entropy RL, which learns the optimal policy via imitating the human behavior.

Concretely, we have the following maximization problem:

$$\max_{\pi} \mathbb{E}_{(s,a)\sim d^{\pi}}[r(s,a) + \alpha\mathcal{H}(\pi(\cdot \mid s))].$$

Here $d^{\pi}$ is the visitation distribution of state $(s,a)$ when rolling out policy $\pi$, i.e. $d^{\pi}(s,a) = \sum_{h=0}^{\infty} \gamma^h \rho_h(s)\pi(a \mid s)$, and $\rho_h$ represents the probability of visiting state $s$ at time step $h$. From the classical result of max-entropy RL, the optimal policy is

$$\pi^{\star}(a \mid s) = \frac{\exp(Q^{\star}(s,a)/\alpha)}{\sum_a \exp(Q^{\star}(s,a)/\alpha)},$$

where the $Q^{\star}$ satisfies the soft Bellman-equation:

$$Q^{\star}(s,a) = r(s,a) + \gamma\mathbb{E}_{s'\sim\mathbb{P}(\cdot\mid s,a)}\left[\alpha\log\left(\sum_{a'}\exp(Q^{\star}(s',a')/\alpha)\right)\right].$$

And the soft $V$ function can be written as

$$V_{\alpha}^{\star}(s) = \alpha\log\left(\sum_{a'}\exp(Q^{\star}(s',a')/\alpha)\right).$$

We can also establish the following relationship between the policy and the soft advantage.

$$\log\pi_{\alpha}(a \mid s) = Q(s,a) - V_{\alpha}(s)/\alpha. \tag{10}$$

When taking $\alpha = 1$, we can see that the log probability is indeed the soft-advantage. This intuition provides a guideline on our algorithm design in later sections.

During the pre-training, we observe samples $\{(s_i,a_i)\}_{i=1}^n \sim \rho(s) \times \pi^{\star}(a \mid s)$. We then approximate the $Q^{\star}$ via minimizing the following cross-entropy loss:

$$\hat{\pi} = \arg\min_{\pi} -\sum_{i=1}^n \log(\pi(a_i \mid s_i)),$$

$$\text{where } \pi \text{ is parameterized as } \hat{\pi}(a \mid s) = \frac{\exp(\hat{Q}(s,a))}{\sum_a \exp(Q(s,a))}.$$

When $N \to \infty$ and $\rho(s)$ is supported everywhere, the policy $\hat{\pi}$ converges to $\pi^{\star}$.

## C.2 SOFT $Q$-BASED FINE-TUNING

Now assume that we have obtained $Q_1^{\star}$ from the pre-training stage, which is the solution to the following reward maximization problem:

$$\max_{\pi} \mathbb{E}_{(s,a)\sim d^{\pi}}[r_1(s,a) + \alpha\mathcal{H}(\pi(\cdot \mid s))].$$

Here $r_1$ is the reward during pre-training. This means that we already know a policy $\pi_1^{\star}$ and a $Q_1^{\star}$ that satisfies

$$\pi_1^{\star}(a \mid s) = \frac{\exp(Q_1^{\star}(s,a)/\alpha)}{\sum_a \exp(Q_1^{\star}(s,a)/\alpha)},$$

$$Q_1^{\star}(s,a) = r_1(s,a) + \gamma\mathbb{E}_{s'\sim\mathbb{P}(\cdot\mid s,a)}\left[\alpha\log\left(\sum_{a'}\exp(Q_1^{\star}(s',a')/\alpha)\right)\right].$$

During fine-tuning, one may target for maximizing a different reward $r_2$ and the reward maximization problem becomes

$$\max_{\pi} \mathbb{E}_{(s,a)\sim d^{\pi}}[r_1(s,a) + r_2(s,a) + \alpha\mathcal{H}(\pi(\cdot \mid s))].$$

In this case, we can fit $r_2(s, a)$ as a reward model from human comparison data. The question is, given $Q_1^\star$ and $r_2$, how shall we find the new policy that maximize the above equation?

We know that the solution will be

$$\pi^\star(a \mid s) = \frac{\exp(Q^\star(s, a)/\alpha)}{\sum_a \exp(Q^\star(s, a)/\alpha)},$$

$$Q^\star(s, a) = r_1(s, a) + r_2(s, a) + \gamma \mathbb{E}_{s' \sim \mathbb{P}(\cdot \mid s, a)} \left[\alpha \log \left(\sum_{a'} \exp(Q^\star(s', a')/\alpha)\right)\right].$$

Since we know that $r_1(s, a) = Q_1^\star(s, a) - \gamma \mathbb{E}_{s' \sim \mathbb{P}(\cdot \mid s, a)} [\alpha \log (\sum_{a'} \exp(Q_1^\star(s', a')/\alpha))]$, we can fit $Q_\theta$ by minimizing the following bellman error:

$$\mathcal{L}_\mathcal{D}(Q_\theta) = \sum_{i=1}^n \left(r_1(s_i, a_i) + r_2(s_i, a_i) + \gamma \mathbb{E}_{s' \sim \mathbb{P}(\cdot \mid s_i, a_i)} \left[\alpha \log \left(\sum_{a'} \exp(Q_\theta(s', a')/\alpha)\right)\right] - Q_\theta(s_i, a_i)\right)^2$$

$$= \sum_{i=1}^n \left(Q_1^\star(s_i, a_i) - \gamma \mathbb{E}_{s' \sim \mathbb{P}(\cdot \mid s_i, a_i)} \left[\alpha \log \left(\sum_{a'} \exp(Q_1^\star(s', a')/\alpha)\right)\right] + r_2(s_i, a_i) \right.$$

$$\left. + \gamma \mathbb{E}_{s' \sim \mathbb{P}(\cdot \mid s_i, a_i)} \left[\alpha \log \left(\sum_{a'} \exp(Q_\theta(s', a')/\alpha)\right)\right] - Q_\theta(s_i, a_i)\right)^2.$$

Then output

$$\pi_\theta(a \mid s) = \frac{\exp(Q_\theta(s, a)/\alpha)}{\sum_{a'} \exp(Q_\theta(s, a')/\alpha)}. \tag{11}$$

# D  ADDITIONAL EXPERIMENTS

## D.1  RESULTS ON THE HH DATASET

In this dataset, each item is comprised of a prompt, a chosen response and a rejected response labeled by human to evaluate the helpfulness and harmlessness of the responses. For the reward model, we use the proxy reward model `Dahoas/gptj-rm-static`[7] with 6B parameters trained from the same dataset based on `EleutherAI/gpt-j-6b`.[8] For all three algorithms, we run two epochs of update after generating 64 responses from randomly sampled prompts. For the 125M model, we use batch size 8 and learning rate $8 \times 10^{-6}$. For the 1B model, we use batch size 2 and learning rate $10^{-6}$. For the 6B and larger models, we use batch size 1 and learning rate $10^{-6}$. We use a 32GB Nvidia V100 GPU for fine-tuning 125M and 1B models, and a 64GB AMD Mi200 GPU for fine-tuning the 6B and larger models. The maximum response length is set to be 128 tokens, and the maximum total sequence length is set to be 1024 tokens. We unfreeze the last two layers during fine-tuning. For each experiment, we run 20k steps in total. The results are plotted as below. In the left of Figure 4, we compare the three methods on the HH dataset. For all three models, we repeat the experiments with three random seeds $0, 100, 1000$, and plot their min, mean and max. We see that with the same amount of data, APA is able to achieve the highest reward in all three cases. We also observe that PPO becomes more stable with large models, potentially due to smaller batch size, or the ability of getting higher reward with a smaller deviation in KL divergence.

On the right of Figure 4, we show how the KL divergence between the current policy and the initial policy changes as a function of the training process for the three seeds. We can see that for all three models, APA provides similar or better KL control than PPO and AWR, although we note that for the 6B model the KL control for PPO is slightly better than APA. Combined with the left part of the figure, we can see that APA is more KL-efficient than PPO and AWR; i.e., it attains a better performance on the reward model under the same KL divergence.

---

[7]`https://huggingface.co/Dahoas/gptj-rm-static`
[8]`https://huggingface.co/EleutherAI/gpt-j-6b`

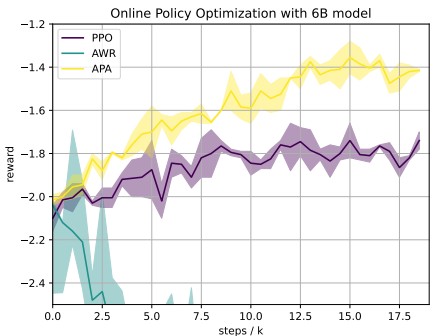 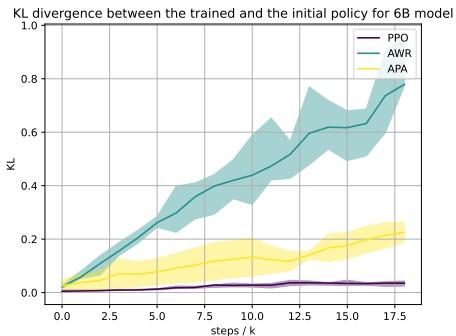

Figure 4: Comparison of the performance of three methods on the HH dataset. Left: The $x$-axis represents the total steps, which are proportional to the amount of data used in the training procedure. The $y$-axis is the reward evaluated by the same reward model. Right: The $x$-axis represents the total steps. The $y$-axis is the KL divergence between the trained model and the initial model.

We include more experiment results in Appendix D, where we fine tune `databricks/dolly-v2-7b`[9] on the same HH dataset, and 2.7B and 6B models on the TLDR dataset[10] for the summarization task.

We also conduct ablation studies on the effect of the adaptive KL controller on PPO and the effect of different choices of $\lambda$ for AWR; see Appendix E. We show in Appendix E.2 that without KL control, PPO can be as sample efficient as APA, but less KL-efficient. We also observe instability even without the KL controller. On the other hand, we observe that changing $\lambda$ provides a straightforward tradeoff between KL control and performance in APA.

## D.2 RESULTS ON THE TLDR DATASET

We fine-tune the `EleutherAI/gpt-neo-2.7B`[11] and 6B `CarperAIopenai_summarize_tldr_sft`[12] models on the TLDR dataset[13] for the summarization task. For `EleutherAI/gpt-neo-2.7B`, we first fine-tune it with supervised fine-tuning on the labeled response in the same summarization dataset, and run RLHF on the supervised fine-tuned policy. The 6B model `CarperAIopenai_summarize_tldr_sft` has already gone through the supervised fine-tuning stage. The reward model is a pre-trained `EleutherAI/gpt-j-6b`[14] reward model for summarization dataset `CarperAI/openai_summarize_comparisons`[15].

We follow the default setting in trlX with seed 0 and 100, and plot the results in Figure 5. One can see that APA is more sample efficient and provides better KL control than PPO in both 2.7B and 6B models.

## D.3 RESULTS ON THE DOLLY MODEL

We fine-tune the `databricks/ dolly-v2-7b`[16] model on the HH dataset. We follow the default setting in trlX with seed 0 and 100, and plot the results in Figure 6. We only include the results for APA and PPO since AWR drops directly. Different from all other experiments, here for APA we set $\lambda = 1$ rather than 0.1 to stablize the training and impose stronger KL control. One can see that APA can still improve over the original dolly 7B model and provide better KL control, while PPO fails to bring further improvement.

---

[9] https://huggingface.co/databricks/dolly-v2-7b
[10] https://huggingface.co/datasets/CarperAI/openai_summarize_comparisons
[11] https://huggingface.co/EleutherAI/gpt-neo-2.7B
[12] https://huggingface.co/CarperAI/openai_summarize_tldr_sft
[13] https://huggingface.co/datasets/CarperAI/openai_summarize_comparisons
[14] https://huggingface.co/EleutherAI/gpt-j-6b
[15] https://huggingface.co/datasets/CarperAI/openai_summarize_comparisons
[16] https://huggingface.co/databricks/dolly-v2-7b

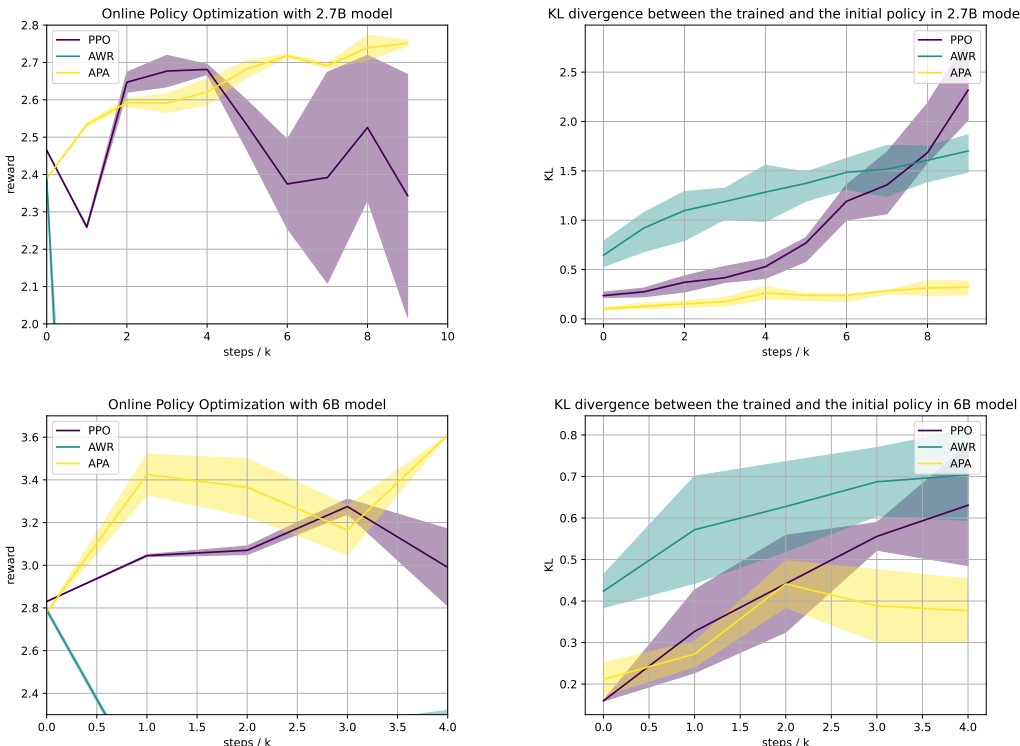

Figure 5: Comparisons of the performance on TLDR dataset. Left: The $x$-axis represents the total steps, which are proportional to the number of data used in the training procedure. The $y$-axis is the reward evaluated by the same reward model. Right: The $x$-axis represents the total steps. The $y$-axis is the KL divergence between the trained model and the initial model.

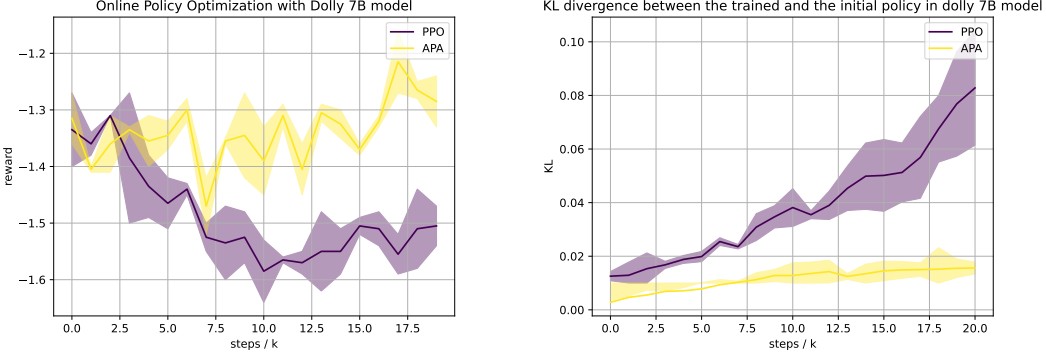

Figure 6: Comparisons of the performance on the dolly 7B model. Left: The $x$-axis represents the total steps, which are proportional to the number of data used in the training procedure. The $y$-axis is the reward evaluated by the same reward model. Right: The $x$-axis represents the total steps. The $y$-axis is the KL divergence between the trained model and the initial model.

### D.4 STACKEXCHANGE EXPERIMENT DETAILS

The hyper-parameters for the Fig. 2 are listed in table 1.

| Parameter | Value |
|---|---|
| Max sequence length | 1024 |
| Max output length | 128 |
| Learning rate | $2 \times 10^{-5}$ |
| Batch size | 16 |
| Gradient Accumulation | 8 |
| SFT LoRA dimension | 16 |
| RM LoRA dimension | 8 |
| RL LoRA dimension | 16 |
| Adaptive KL initial KL coeff (PPO) | 0.1 |
| Adaptive KL target KL coeff (PPO) | 6 |
| $\lambda$ (APA) | 0.1 |

Table 1: Hyper-parameters for StackExchange experiments as shown in Fig. 2

### D.5 RESULTS ON MUJOCO

We now demonstrate the performance of our proposed algorithm APA on continuous control tasks, Mujoco (Todorov et al., 2012). We used the Spinning Up[17] framework for our experimentation. The policy is represented by a fully-connected multilayer perceptron with two hidden layers of 64 units, and tanh nonlinearities, outputting the mean of a Gaussian distribution, with variable standard deviations same as in (Schulman et al., 2017). We tested on four environments: HalfCheetah-v2, Humanoid-v2, Hopper-v2 and Ant-v2. Fig. 7 shows that APA outperforms PPO in three of them. This indicates that APA is more general algorithm which can be applied to conventional RL tasks to language models. We optimized the hyper-parameters for the best performance of both algorithms. The final hyper-parameters for these experiments are listed in table 2.

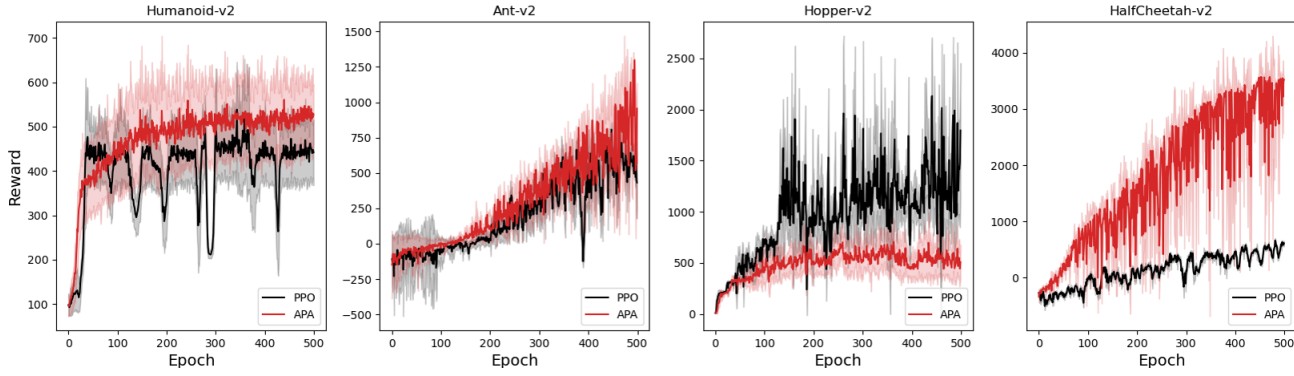

Figure 7: Comparison of the performance of APA and PPO on Mujoco environments

| Parameter | Value |
|---|---|
| $\gamma$ | 0.99 |
| Clip ratio (PPO) | 0.2 |
| GAE parameter (PPO) | 0.97 |
| KL coeff (PPO) | 1 |
| $\lambda$ (APA) | 0.1 |
| Learning rate | $3 \times 10^{-4}$ |

Table 2: Hyper-parameters for continuous control Mujoco tasks

---

[17]https://spinningup.openai.com/en/latest/

# E  ABLATION STUDIES

## E.1  KL CONTROL IN APA

In this section, we show how the performance and KL divergence change with different values of $\lambda$. We set $\lambda = 0.1, 1$ for the 125M model and plot their performances in Figure 8 with seed 1000. One can see that the choice of $\lambda$ directly determines the level of KL control, along with the convergent point APA reaches. This shows that $\lambda$ provides a clear trade-off between KL control and model performance.

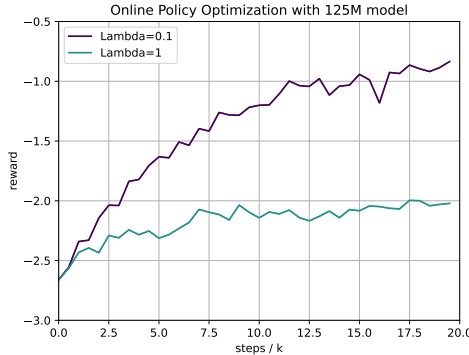 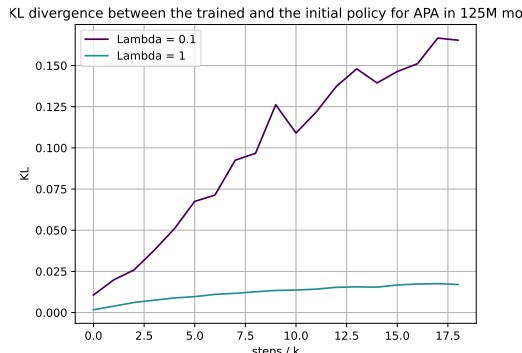

Figure 8: Comparisons of the performance between different $\lambda$ on the 125M model. Left: The $x$-axis represents the total steps, which are proportional to the number of data used in the training procedure. The $y$-axis is the reward evaluated by the same reward model. Right: The $x$-axis represents the total steps. The $y$-axis is the KL divergence between the trained model and the initial model.

## E.2  KL CONTROL IN PPO

We show how the performance and KL divergence change with or without adaptive KL control in PPO. We plot their performances in Figure 9 for 125M model with seed 1000. For PPO with adaptive KL controller, the initial KL coefficient is set to be 0.05. One can see that without KL control, PPO converges to a higher reward compared to APA in Figure 8, at a cost of a significantly higher KL divergence. On the other hand, the reward of PPO with adaptive KL control begins to drop in the middle. This is due to the large deviation from the original policy, which leads to a much larger KL regularization term that dominates the reward. Compared with Figure 8, one can see that APA provides more stable and controllable KL regularization.

## E.3  EXPERIMENTS FOR OFFLINE LEARNING

We conduct experiments for offline learning as well. The offline dataset is selected to be all the prompts and responses from the HH dataset, with reward labeled by the reward model. We use the trained GPT-J reward function to label the reward for all the offline data, and compare ILQL, AWR and APA on the same 125M and 1B model after supervised fine-tuning with seed 1000. The result is given in Figure 10. From the results, one can see that AWR performs better than ILQL, and APA cannot be directly adapted to the offline case. Furthermore, offline learning cannot help too much after the supervised fine-tuning stage, potentially due to the large distribution shift between the offline data and the current policy.

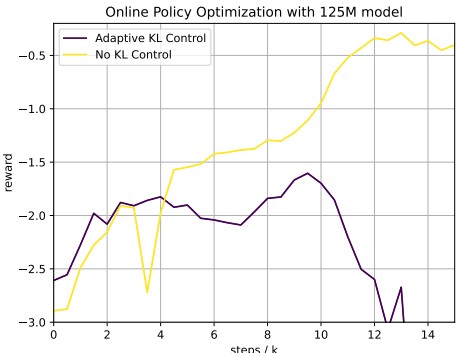 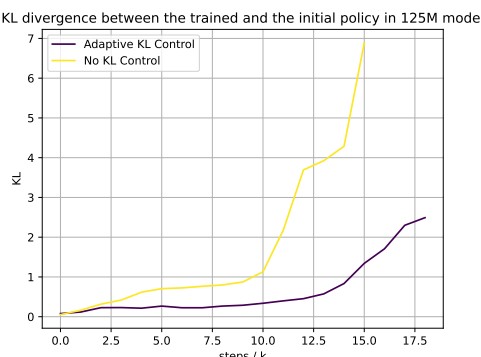

Figure 9: Comparisons of the performance of PPO on the 125M model. Left: The $x$-axis represents the total steps, which are proportional to the number of data used in the training procedure. The $y$-axis is the reward evaluated by the same reward model. Right: The $x$-axis represents the total steps. The $y$-axis is the KL divergence between the trained model and the initial model.

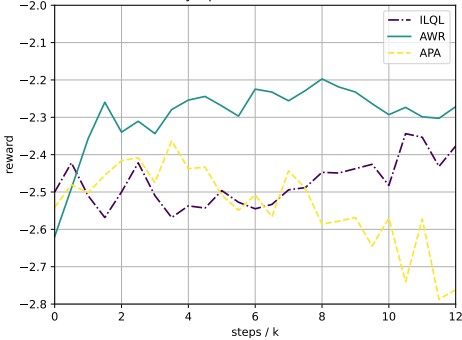 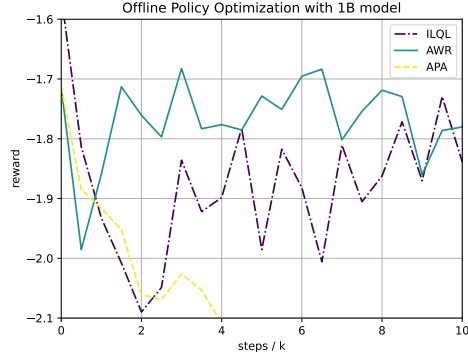

Figure 10: Comparisons of the performance between ILQL, AWR and APA on the offline learning dataset.

## F   PROOF OF THEOREM 1

*Proof.* From the well-specified assumption $\pi^\star \in \{\pi_\theta \mid \theta \in \Theta\}$, we know that there exists some $\theta^\star \in \Theta$ such that $\pi_{\theta^\star} = \pi^\star$. For the population loss, we know that

$$
\begin{aligned}
\mathcal{L}^{\mathsf{APA}}(\theta^\star) &= \mathbb{E}_{(s,a)\sim d_{s,a}^{\pi_{\mathsf{old}}}}\left[\left(\log \pi_{\theta^\star}(a \mid s) - \mathsf{Adv}(s,a)/\lambda - \log \pi_{\mathsf{init}}(a \mid s)\right)^2\right] \\
&= \mathbb{E}_{(s,a)\sim d_{s,a}^{\pi_{\mathsf{old}}}}\left[\left(\log \pi^\star(a \mid s) - \mathsf{Adv}(s,a)/\lambda - \log \pi_{\mathsf{init}}(a \mid s)\right)^2\right] \\
&= 0.
\end{aligned}
$$

Thus for any $\theta' \in \arg\min_{\theta\in\Theta} \mathcal{L}^{\mathsf{APA}}(\theta)$, there must be $\mathcal{L}^{\mathsf{APA}}(\theta') = 0$, which is equivalent to

$$
\mathbb{E}_{(s,a)\sim d_{s,a}^{\pi_{\mathsf{old}}}}\left[\left(\log \pi_{\theta'}(a \mid s) - \mathsf{Adv}(s,a)/\lambda - \log \pi_{\mathsf{init}}(a \mid s)\right)^2\right] = 0.
$$

This means that for any $s, a$ on the support of $d_{s,a}^{\pi_{\mathsf{old}}}$, we have $\pi_{\theta'}(a \mid s) = \pi^\star(a \mid s)$.

For the second part of the theorem, we know from Hoeffding's inequality that for any fixed $\theta \in \Theta$,

$$
\begin{aligned}
|\mathcal{L}^{\mathsf{APA}}(\theta) - \widehat{\mathcal{L}}^{\mathsf{APA}}(\theta; \mathcal{D})| =& \left|\frac{1}{n}\sum_{i=1}^{n}\left(\log \pi_\theta(a_i \mid s_i) - \mathsf{Adv}(s_i,a_i)/\lambda - \log \pi_{\mathsf{init}}(a_i \mid s_i)\right)^2 \right.\\
&\left. - \mathbb{E}\left[\left(\log \pi_\theta(a \mid s) - \mathsf{Adv}(s,a)/\lambda - \log \pi_{\mathsf{init}}(a \mid s)\right)^2\right]\right| \\
\leq& C \cdot (B_2/\lambda - 2\log(B_1))^2 \sqrt{\frac{\log(1/\delta)}{n}}.
\end{aligned}
$$

Let the $\Theta_\epsilon$ be $\epsilon$-covering of $\Theta$ under $\ell_2$ norm, i.e. for any $\theta \in \Theta$, one can find some $\theta' \in \Theta_\epsilon$ such that $\|\theta - \theta'\|_2 \leq \epsilon$. We also have $|\Theta_\epsilon| \leq (1/\epsilon)^d$. By taking union bound, we know that for all $\theta \in \Theta_\epsilon$, with probability at least $1 - \delta$,

$$
|\mathcal{L}^{\mathsf{APA}}(\theta) - \widehat{\mathcal{L}}^{\mathsf{APA}}(\theta; \mathcal{D})| \leq C \cdot (B_2/\lambda - \log(B_1))^2 \sqrt{\frac{d\log(1/(\epsilon\delta))}{n}}. \tag{12}
$$

Let $\hat{\theta}$ be the minimizer of $\widehat{\mathcal{L}}^{\mathsf{APA}}(\theta; \mathcal{D})$. Then we know that there exists some $\hat{\theta}_\epsilon \in \Theta_\epsilon$ such that $\|\hat{\theta} - \hat{\theta}_\epsilon\| \leq \epsilon$. This further implies that

$$
\begin{aligned}
&|\mathcal{L}^{\mathsf{APA}}(\hat{\theta}) - \mathcal{L}^{\mathsf{APA}}(\hat{\theta}_\epsilon)| \\
=&|\mathbb{E}\left[\left(\log \pi_{\hat{\theta}}(a \mid s) - \mathsf{Adv}(s,a)/\lambda - \log \pi_{\mathsf{init}}(a \mid s)\right)^2\right] - \mathbb{E}\left[\left(\log \pi_{\hat{\theta}_\epsilon}(a \mid s) - \mathsf{Adv}(s,a)/\lambda - \log \pi_{\mathsf{init}}(a \mid s)\right)^2\right]| \\
\leq&C(B_2/\lambda - \log(B_1))L\epsilon. \tag{13}
\end{aligned}
$$

Similarly, we also have $|\widehat{\mathcal{L}}^{\mathsf{APA}}(\hat{\theta}) - \widehat{\mathcal{L}}^{\mathsf{APA}}(\hat{\theta}_\epsilon)| \leq C(B_2/\lambda - \log(B_1))L\epsilon$. Overall, we have

$$
\mathcal{L}^{\mathsf{APA}}(\hat{\theta}) = (\mathcal{L}^{\mathsf{APA}}(\hat{\theta}) - \mathcal{L}^{\mathsf{APA}}(\hat{\theta}_\epsilon)) + (\mathcal{L}^{\mathsf{APA}}(\hat{\theta}_\epsilon) - \widehat{\mathcal{L}}^{\mathsf{APA}}(\hat{\theta}_\epsilon)) + (\widehat{\mathcal{L}}^{\mathsf{APA}}(\hat{\theta}_\epsilon) - \widehat{\mathcal{L}}^{\mathsf{APA}}(\hat{\theta})) + \widehat{\mathcal{L}}^{\mathsf{APA}}(\hat{\theta}).
$$

For the first and third difference, from Equation (13) we know that they are both bounded by $C(B_2/\lambda - \log(B_1))L\epsilon$. For the second difference, we know from Equation (12) that it is bounded by $C(B_2/\lambda - \log(B_1))^2 \sqrt{\frac{d\log(1/(\epsilon\delta))}{n}}$. Lastly, we know that $\widehat{\mathcal{L}}^{\mathsf{APA}}(\hat{\theta}) = 0$ since $\hat{\theta} \in \arg\min_\theta \widehat{\mathcal{L}}^{\mathsf{APA}}(\theta)$ and $\widehat{\mathcal{L}}^{\mathsf{APA}}(\theta^\star) = 0$. Thus overall, we have

$$
\mathcal{L}^{\mathsf{APA}}(\hat{\theta}) \leq C\left((B_2/\lambda - \log(B_1))L\epsilon + (B_2/\lambda - \log(B_1))^2 \sqrt{\frac{d\log(1/(\epsilon\delta))}{n}}\right).
$$

Taking $\epsilon = 1/(Ln)$ finishes the proof.     $\square$

