# OpenReview forum: "Fine-Tuning Language Models with Advantage-Induced Policy Alignment"
_ICLR.cc/2024/Conference — Submitted to ICLR 2024_

### Official Review · Reviewer_59is · 2023-10-29

**Soundness:** 2 fair
**Presentation:** 3 good
**Contribution:** 2 fair
**Rating:** 5
**Confidence:** 3

**Summary:**

This paper introduces Advantage-Induced Policy Alignment (APA), a method that applies advantage weighted regression (AWR) to Language Models (LLMs) alignment tasks and replaces the KL-divergence with a squared error loss function.

**Strengths:**

The paper is well-written. The analysis and discussion are insightful and contribute to a better understanding of the proposed method.
Experiments on multiple datasets and LLMs demonstrate significant improvements over AWR.

**Weaknesses:**

The primary concern lies in the novelty and motivation of the proposed method. It seems that the objective represents a minor adjustment to previous work, where KL divergence (p log(p/q)) is replaced by a new f-divergence (p log^2(p/q)). The motivation behind this change remains unclear. The authors claim that APA no longer requires estimating the importance ratio, but it is worth noting that equation (9), which contains log(π_theta / π_init), still necessitates calculating this ratio. The authors have pointed out discrepancies between the original target and the final loss of AWR as well as the changing of π_old in the online case as motivating factors for APA. However, it appears that APA still faces these issues. Another potential motivation could be sample efficiency, but the authors have not explained why using the new f-divergence is more sample-efficient.

## Theoretical Results:
The authors claim to provide a comparison among APA, PPO, and AWR theoretically in the introduction, but only the upper bound of APA is presented in Theorem 1. This leaves readers unable to derive the theoretical advantages of the proposed method. Additionally, the assumption for explaining why Z(s) approximates to 1 is deemed overly strong.

## Experiments:
Given that both APA and DPO [1] are derived from the optimal policy and are non-RL methods, it is suggested that APA should be compared with more existing works, such as DPO.

Recent research [2] has highlighted the significance of considering various divergences, each offering its unique balance between alignment and diversity trade-offs. Given that the core contribution of this paper revolves around the introduction of a novel f-divergence for distribution matching, it is imperative that the authors conduct a comparative analysis by pitting the new f-divergence against commonly used f-divergences, such as reverse KL, total variation (TV), and Jensen-Shannon (JS) divergences.
Figure 2 suggests that PPO may not have converged due to a large KL penalty.

## Typos:
In the formation of the PPO loss, \hat{Adv} should be "adv."
[1] Rafailov R, Sharma A, Mitchell E, et al. "Direct preference optimization: Your language model is secretly a reward model." ICML Workshop MFPL, 2023.
[2] Go D, Korbak T, Kruszewski G, et al. Aligning language models with preferences through f-divergence minimization[J]. International Conference on Machine Learning. PMLR, 2023.

**Questions:**

Recent research [3] argues that the KL penalty can be removed, and the authors themselves find that PPO without KL penalty converges to a higher reward compared to APA, as shown in Figure 7. Does this imply that PPO outperforms APA when using smaller or no KL penalty terms? It is important to note that the reviewer does not believe that the higher cost of KL divergence is the issue.

[3] Gao L, Schulman J, Hilton J. "Scaling laws for reward model overoptimization." International Conference on Machine Learning. PMLR, 2023: 10835-10866.

---

> ### Author Response · Authors · 2023-11-22
>
> ## Comment 1
>
> **Reviewer:**
>
> > The primary concern lies in the novelty and motivation of the proposed method. It seems that the objective represents a minor adjustment to previous work, where KL divergence (p log(p/q)) is replaced by a new f-divergence (p log^2(p/q)). The motivation behind this change remains unclear. The authors claim that APA no longer requires estimating the importance ratio, but it is worth noting that equation (9), which contains log(π_theta / π_init), still necessitates calculating this ratio. The authors have pointed out discrepancies between the original target and the final loss of AWR as well as the changing of π_old in the online case as motivating factors for APA. However, it appears that APA still faces these issues. Another potential motivation could be sample efficiency, but the authors have not explained why using the new f-divergence is more sample-efficient.
>
>
> **Response:**
>
> Thank you for your comments. We would like to clarify first that APA does not face the same issue introduced by changing $\pi_{old}$ as AWR. Intuitively, this is because minimizing $\pi_{old} \log(\pi / \pi_{init})$ (actual implementation of AWR) is different from minimizing $\pi_{init} \log(\pi / \pi_{init})$ (the ideal target of AWR), while minimizing  $\pi_{old} (\pi - \pi_{init})^2$ (actual implementation of APA) is similar to minimizing $\pi_{init} (\pi - \pi_{init})^2$ (the ideal target of APA) as long as the policy $\pi_{old}$ and $\pi_{init}$ covers enough state-action pairs. This benefit is naturally introduced by replacing KL with squared loss, and is our main motivation for designing APA. We have also provided theoretical guarantees to formalize this intuition.
>
>
>
> ## Comment 2
>
> **Reviewer:**
>
>
> > The authors claim to provide a comparison among APA, PPO, and AWR theoretically in the introduction, but only the upper bound of APA is presented in Theorem 1. This leaves readers unable to derive the theoretical advantages of the proposed method. Additionally, the assumption for explaining why Z(s) approximates to 1 is deemed overly strong.
>
>
> **Response:**
>
> Thank you for the comment. We discuss in Appendix A why Z(s) can be approximated as 1, which is validated in experiments. This assumption is also used implicitly in the derivation of AWR.
>
> In terms of theoretical guarantees for PPO and AWR, it can be hard to provide bounds due to their complicated nature, especially for PPO which relies on estimating both value, advantage and updating the model. For AWR, we show in Equation (7) that it converges to the wrong target that is inconsistent with Equation (3). Thus theoretically, we know that AWR is not convergent while APA is convergent.
>
>
>
>
> ## Comment 3
>
> **Reviewer:**
>
> > Given that both APA and DPO [1] are derived from the optimal policy and are non-RL methods, it is suggested that APA should be compared with more existing works, such as DPO.
>
> **Response:**
>
> Thank you for the comment. In fact, we still view APA as an RL alternative to PPO. To illustrate our point, we have included comparison between APA and PPO on the traditional mujoco task in Appendix D.3, which DPO is not able to accompolish since DPO requires comparison data.
>
>
> ## Comment 4
>
> **Reviewer:**
>
>
> > Recent research [2] has highlighted the significance of considering various divergences, each offering its unique balance between alignment and diversity trade-offs. Given that the core contribution of this paper revolves around the introduction of a novel f-divergence for distribution matching, it is imperative that the authors conduct a comparative analysis by pitting the new f-divergence against commonly used f-divergences, such as reverse KL, total variation (TV), and Jensen-Shannon (JS) divergences. Figure 2 suggests that PPO may not have converged due to a large KL penalty.
>
> **Response:**
>
> Thank you for your comments. We choose to mainly focus on KL in Equation (3) since that is a traditional divergence where both APA and PPO rely on. We are happy to include more comparisons on different f-divergences in the final version of the paper.
>
> In Figure 2, we control the KL divergence of APA and PPO to be on the same level to ensure a fair comparison. In this case, we can observe that APA is more KL-efficient and sample-efficient than PPO.
>
> [1] Rafailov R, Sharma A, Mitchell E, et al. "Direct preference optimization: Your language model is secretly a reward model." ICML Workshop MFPL, 2023.
>
> [2] Go D, Korbak T, Kruszewski G, et al. Aligning language models with preferences through f-divergence minimization[J]. International Conference on Machine Learning. PMLR, 2023.

---

> > ### Author Response · Authors · 2023-11-22
> >
> > ## Comment 5
> >
> > **Reviewer:**
> >
> > > Recent research [3] argues that the KL penalty can be removed, and the authors themselves find that PPO without KL penalty converges to a higher reward compared to APA, as shown in Figure 7. Does this imply that PPO outperforms APA when using smaller or no KL penalty terms? It is important to note that the reviewer does not believe that the higher cost of KL divergence is the issue.
> >
> >
> > **Response:**
> >
> > Thank you for the comment. In fact, the results in [3] suggest that removing KL penalty will not affect the “gold reward-KL frontier”. This suggests that no matter whether we have KL regularization, if KL is much larger, the gold reward will still be worse than the case of small KL. In Appendix, we find that PPO achieves a higher reward than APA, but at the cost of a much higher KL divergence. According to [3], this means that the PPO may have lower gold reward even though the proxy reward is higher.
> >
> > [3] Gao L, Schulman J, Hilton J. "Scaling laws for reward model overoptimization." International Conference on Machine Learning. PMLR, 2023: 10835-10866.

---

> > > ### Comment · Reviewer_59is · 2023-11-23
> > >
> > > Thank you for your response. After carefully reading your reply and the comments from other reviewers, I have decided to keep the score.

---

### Official Review · Reviewer_2o5E · 2023-10-30

**Soundness:** 2 fair
**Presentation:** 4 excellent
**Contribution:** 2 fair
**Rating:** 3
**Confidence:** 4

**Summary:**

This paper proposes an alternative approach to the PPO algorithm for RLHF. The proposed approach has a clear mathematical motivation, where the key idea is to replace the KL-regularizer with a squared-error of the log-probability. Empirical results show that the proposed algorithms consistently outperform PPO in some problems.

**Strengths:**

1 The presentation and intuition is clear, following the theoretical solution of KL-regularized optimization problem with several reasonable modifications.

2 Some of the empirical results consistently demonstrate that the proposed algorithm is promising in terms of stability, sample efficiency, and also reward optimization. The considered datasets are also standard in RLHF literature.

Overall, I feel that the authors have proposed a promising alternative approach to the PPO algorithm. And it would be interesting to see if further industry-level models are aligned with the proposed algorithm in the future. However, the paper is still not ready for publication. I have a couple of questions in the weakness part.

**Weaknesses:**

1 While I can understand the mathematical derivations along the line and it is great to see that the loss of APA is provably convergent, I am curious why the square loss is better than the KL-divergence (it is because of the guarantee provided in theorem 1?). In a more general sense, as the new loss function can be viewed as a different f-divergence, and we know that with a f-divergence as the regularizer, we can also obtain another variants of (3), do these f-divergences work better than KL? I am not sure whether there are studies in the literature of PPO but I do see a work considering this modification in DPO [1]. Can you comment on this or provide more intuitions why the squared error works better than the KL and how does it compare with general f-divergence? I think a more systematic and comprehensive comparisons among all these common f-divergence choices could largely improve the paper.

2 It seems that the modification also applies to the DRL scenarios beyond the LLMs. But as far as I know, the prominent approach in research and industry is still the KL-based PPO. I am wondering whether this is due to the special case of LLMs or this modification could also outperform PPO in classical applications.

3 After looking at the dataset in the huggingface, the HH-RLHF dataset used in this paper indeed only contains the helpful part. This is fine as multi-objective alignments could be more complicated and is not the focus of this paper but I think you may explicitly mention this somewhere.

4 Why do you choose to finetune only the last two layers (on the HH dataset in appendix C)? It seems that with the used GPT-J-6B reward model, we might get much higher reward with PPO [2] (although it does depends on the implementation and starting checkpoint). I am not sure whether this is because you only choose to finetune the last two layers but it seems that in practice, we tends to use low-rank adapter instead of freezing most of the layers.

5 In figure 6, the PPO does not work normally as the reward seems to be decrease from the beginning.  I understand that PPO often drops suddenly but this could be mitigated by using larger KL penalty and also smaller learning rate and less epochs for each iteration. I think the PPO is not tuned to the best performance so far and using the default parameters in trlx is not enough.

A minor typo: I think the state space of the LLM should be the product of the token spaces instead of the union.

[1] Beyond Reverse KL: Generalizing Direct Preference Optimization with Diverse Divergence Constraints

[2] RRHF: Rank Responses to Align Language Models with Human Feedback without tears

**Questions:**

See weakness.

---

> ### Author Response · Authors · 2023-11-22
>
> Thank you for the comments and suggestions. Below is our response.
>
> ## Comment 1
>
> **Reviewer:**
>
> > While I can understand the mathematical derivations along the line and it is great to see that the loss of APA is provably convergent, I am curious why the square loss is better than the KL-divergence (it is because of the guarantee provided in theorem 1?). In a more general sense, as the new loss function can be viewed as a different f-divergence, and we know that with a f-divergence as the regularizer, we can also obtain another variants of (3), do these f-divergences work better than KL? I am not sure whether there are studies in the literature of PPO but I do see a work considering this modification in DPO [1]. Can you comment on this or provide more intuitions why the squared error works better than the KL and how does it compare with general f-divergence? I think a more systematic and comprehensive comparisons among all these common f-divergence choices could largely improve the paper.
>
>
> **Response:**
>
> Thank you for your comments. In fact, there are two places in the derivation where KL divergence is involved:
>
> 1. For the target of policy improvement, we always start with Equation (2) which maximizes expected advantage with KL regularization, and get Equation (3) as a closed-form solution. For both AWR and APA, we always aim at learning the closed-form solution in (3) which is derived with KL regularization.
>
> 2. The closed-form solution in (3) may not be inside the parametrized neural network family, thus we need to project the closed-form solution policy in (3) to the space of the neural network. APA mainly differs from AWR in terms of the choice of projection distances, where we find that squared distance appears to be a much better choice than direct KL divergence projection. The intuition is that when approximating the KL loss with sampled state-action pairs in AWR, it implicitly replaces pi_init with pi_old. Thus AWR is not optimizing for the ideal KL target.
>
> Other forms of f-divergence will lead to a different closed-form target distributions as is suggested in [1]. For those new target distributions, it is still possible to use squared loss or KL as the projection distance. We are happy to include more comprehensive discussions and comparisons in the final version of the paper.
>
>
> ## Comment 2
>
> **Reviewer:**
>
> > It seems that the modification also applies to the DRL scenarios beyond the LLMs. But as far as I know, the prominent approach in research and industry is still the KL-based PPO. I am wondering whether this is due to the special case of LLMs or this modification could also outperform PPO in classical applications.
>
>
> **Response:**
>
> Thank you for your comments. We have included in Appendix D.5 on the experiments in 4 traditional RL tasks in Mujoco, where we observe that APA outperforms PPO for 3 of them. This shows the potential of APA being applied to traditional RL tasks as well.
>
>
> ## Comment 3
>
> **Reviewer:**
>
> > After looking at the dataset in the huggingface, the HH-RLHF dataset used in this paper indeed only contains the helpful part. This is fine as multi-objective alignments could be more complicated and is not the focus of this paper but I think you may explicitly mention this somewhere.
>
> **Response:**
>
> Thank you for your comments. We have revised the paper and mention this point in the main text.
>
>
> ## Comment 4
>
> **Reviewer:**
>
> > Why do you choose to finetune only the last two layers (on the HH dataset in appendix C)? It seems that with the used GPT-J-6B reward model, we might get much higher reward with PPO [2] (although it does depends on the implementation and starting checkpoint). I am not sure whether this is because you only choose to finetune the last two layers but it seems that in practice, we tends to use low-rank adapter instead of freezing most of the layers.
>
> **Response:**
>
> Thank you for your comments. In fact, we use a low-rank adapter for our stack-exchange dataset fine-tuning to ensure a fair comparison with both LoRA and frozen-layer-based methods. The reward with PPO highly depends on the chosen evaluation dataset.

---

> > ### Author Response · Authors · 2023-11-22
> >
> > ## Comment 5
> >
> > **Reviewer:**
> >
> > > In figure 6, the PPO does not work normally as the reward seems to decrease from the beginning. I understand that PPO often drops suddenly but this could be mitigated by using larger KL penalty and also smaller learning rate and less epochs for each iteration. I think the PPO is not tuned to the best performance so far and using the default parameters in trlx is not enough.
> >
> >
> > **Response:**
> >
> > Thank you for your comments. The beginning drop of PPO can be normal due to the inaccurate estimation of value functions. After the value function converges or stabilizes, the performance of PPO will begin to increase. We are happy to further fine-tune the performance of PPO in the final version. Althoughwe adopt the default hyperparameters for PPO, we indeed have conducted extensive hyperparameter search for all the three algorithms. In Appendix D, we list the most significant hyperparameter that changes the result, which is the KL control coefficient. We provide results for APA when the KL coefficient is set to be $0.1, 1, 10$, and the results for PPO when the initial KL coefficient is $0$ or $0.05$.  It turns out that if we run PPO without KL control, the reward will always increase, but at a cost of much higher KL divergence. But once we increase the KL coefficient, we observe some decrease in the reward, potentially due to both stringent KL control and the instability of the PPO algorithms. Besides this, we do find that the default hyperparameters for PPO give the most robust experimental results for our settings. However, we hope to mention that one motivation of proposing APA is that there are so many hyperparameters to tune in PPO, while APA only requires the KL coefficient, which is another huge benefit.
> >
> >
> > ## Comment 6
> >
> > **Reviewer:**
> >
> >
> > > A minor typo: I think the state space of the LLM should be the product of the token spaces instead of the union.
> >
> >
> > **Response:**
> >
> > Thank you for your suggestion! In fact, the state space is defined as the union of the product of the token spaces with variable number of tokens. So our first union is union over all possible lengths in the token.

---

> > > ### Comment · Reviewer_2o5E · 2023-11-22
> > > **Thanks for the response and clarification**
> > >
> > > I have read the responses and revised paper.
> > >
> > > I still feel that designing algorithm from the closed-form solution is promising, in view of recent progress of new SFT-based algorithms (iterative finetuning and DPO, for instance). It is interesting to see whether this viewpoint can be also helpful for drl approach. Therefore, the idea of this paper is sound and clever. But there is a lot of works to do before the paper could be finally published. So I keep the score toward reject for this round.

---

### Official Review · Reviewer_qc78 · 2023-10-31

**Soundness:** 4 excellent
**Presentation:** 3 good
**Contribution:** 3 good
**Rating:** 8
**Confidence:** 5

**Summary:**

This work presents Advantage Induced Policy-Alignment (APA) for language model finetuning. This work first identifies a source of instability and optimization target mismatch in Advantage Weighted Regression (AWR) and addresses it with APA by minimizing a different f-divergence rather than the KL. They show on multiple RLHF tasks that APA leads to more stable and sample-efficient learning when compared to online RL (PPO) and AWR.

**Strengths:**

- Good presentation of the idea in a clear way. The paper is well written and easy to follow.
- Experimentation. Multiple tasks are evaluated with a good discussion/comparison between the RLHF tradeoff of KL from SFT model as well as reward optimization.

**Weaknesses:**

Minor comments:
- In the main text, there is a reference to a connection to soft-q learning but it seems like only the f-divergence interpretation is discussed.
- For the 125M parameter experiments, it seems like PPO is not properly tuned to optimize for the reward.

**Questions:**

How important was it to estimate a good critic? Was there anything else that was needed to learn a better A rather than having a separate model? This seems to be an interesting angle to experiment some more since the provable guarantees also relies on having a good advantage function for $\pi_{old}$.

---

> ### Author Response · Authors · 2023-11-22
>
> Thank you for the comments and suggestions. Below is our response.
>
> ## Comment 1
>
> **Reviewer:**
>
> > In the main text, there is a reference to a connection to soft-q learning but it seems like only the f-divergence interpretation is discussed.
>
>
> **Response:**
>
> We have included a new Appendix C discussing an alternative loss inspired by soft-Q learning. In fact, the soft-Q learning loss is our first attempt before APA. Unfortunately, it does not give as good performance as APA or PPO. So we only provide intuitions and discussions in the Appendix. We have also updated the main text to connect to the Appendix better.
>
> ## Comment 2
>
> **Reviewer:**
>
> > For the 125M parameter experiments, it seems like PPO is not properly tuned to optimize for the reward.
>
>
> **Response:**
>
> We have optimized the PPO performance as hard as we can. In fact, it is interesting to see that with a larger language model (3B or 6B), the performance of PPO seems to be more stabilized than that of 125M parameters. We have also conducted ablation study in Appendix D for the effect of different hyperparameters for PPO and APA algorithm.
>
> ## Comment 3
>
> **Reviewer:**
>
> > How important was it to estimate a good critic? Was there anything else that was needed to learn a better A rather than having a separate model? This seems to be an interesting angle to experiment some more since the provable guarantees also relies on having a good advantage function for pi_old.
>
> **Response:**
>
> It is very important to learn a good critic / advantage function, which is required for both PPO and APA. In fact, during our experiment, the value function converges very fast, which is a good sign that the advantage might be estimated very well due to the same advantage estimation technique used in PPO.

---

### Official Review · Reviewer_GAYy · 2023-11-04

**Soundness:** 2 fair
**Presentation:** 2 fair
**Contribution:** 2 fair
**Rating:** 5
**Confidence:** 4

**Summary:**

This paper proposes a new algorithm for fine-tuning language models with RLHF. The algorithm, APA, uses a squared error loss function that aligns the output policy of the language model with a target policy based on the estimated advantages. The authors claim that APA has advantages over existing RLHF methods, such as PPO and AWR, in terms of sample efficiency, stability, and hyperparameter sensitivity. Besides, the paper provides theoretical justification for APA. The proposed algorithm is evaluated on two language tasks: StackExchange question answering and HH dataset, and outperforms existing methods such as PPO and AWR in terms of sample efficiency, stability, and KL control.

**Strengths:**

The research objective is clear and the paper is well-motivated. I do notice that PPO is sometimes unstable for training language models, and the model performance may drop with the training goes on. The proposed algorithm is simple and seems effective on the evaluation tasks. I appreciate the authors' effort in providing theoretical justification for the design of the loss function and the convergence of the algorithm.

**Weaknesses:**

The major reason I tend to reject is the scope of evaluate tasks. As RLHF (with PPO) has been well verified on ChatGPT, which has great generalization ability, PPO can be utilized to train language models in scale. In addition to training language models, PPO has stand the test of time in many other area such as robotics control. I believe that is why researchers use PPO to fine-tune many large language models. However, the evaluation tasks in this paper locate in a specific domain. I think it is not enough to be an alternative to PPO.

Besides, I also have some minor concerns:
1. The paper does not provide much discussion on the choice of the hyperparameter $\lambda$, which controls the trade-off between the expected advantage and the KL divergence. Besides, as this paper proposes an optimization algorithm, the ablation study is required.
2. There seems lacking an analyze the qualitative differences between the outputs of different algorithms, or the potential harms or biases of the reward models.

**Questions:**

My major concerns have been listed in 'Weakness' section. Here are some other questions:

1. Can APA generalize to other language tasks or domains?
2. What do you think are important for a RL algorithm in RLHF training?

---

> ### Author Response · Authors · 2023-11-22
>
> Thank you for the comments and suggestions. Below is our response.
>
> ## Comment 1
>
> **Reviewer:**
>
> > The major reason I tend to reject is the scope of evaluating tasks. As RLHF (with PPO) has been well verified on ChatGPT, which has great generalization ability, PPO can be utilized to train language models in scale. In addition to training language models, PPO has stood the test of time in many other areas such as robotics control. I believe that is why researchers use PPO to fine-tune many large language models. However, the evaluation tasks in this paper are located in a specific domain. I think it is not enough to be an alternative to PPO. Can APA generalize to other language tasks or domains?
>
>
>  **Response:**
>
> Thank you for your comments. We have included in Appendix D.5 of the updated draft on the experiments in 4 traditional RL tasks in Mujoco, where we observe that APA outperforms PPO for 3 of them. This shows the potential of APA being applied to traditional RL tasks as well.
>
>
> ## Comment 2
>
> **Reviewer:**
>
> > The paper does not provide much discussion on the choice of the hyperparameter $\lambda$, which controls the trade-off between the expected advantage and the KL divergence. Besides, as this paper proposes an optimization algorithm, the ablation study is required.
>
>
>
>  **Response:**
>
> Thank you for your comments. We have ablation study included in Appendix D. In Appendix D, we list the most significant hyperparameter that changes the result, which is the KL control coefficient. We provide results for APA when the KL coefficient is set to be $0.1, 1, 10$, and the results for PPO when the initial KL coefficient is $0$ or $0.05$.  It turns out that if we run PPO without KL control, the reward will always increase, but at a cost of much higher KL divergence. But once we increase the KL coefficient, we observe some decrease in the reward, potentially due to both stringent KL control and the instability of the PPO algorithms. Besides this, we do find that the default hyperparameters for PPO give the most robust experimental results for our settings. However, we hope to mention that one motivation of proposing APA is that there are so many hyperparameters to tune in PPO, while APA only requires the KL coefficient, which is another huge benefit.
>
>
> ## Comment 3
>
> **Reviewer:**
>
> > There seems lacking an analyze the qualitative differences between the outputs of different algorithms, or the potential harms or biases of the reward models.
>
>
>  **Response:**
>
> Thank you for your comments. We are happy to include some output examples in the final version, along with discussions on quantitative differences and potential harms.
>
>
>
> ## Comment 4
>
> **Reviewer:**
>
> > What do you think are important for a RL algorithm in RLHF training?
>
>
> **Response:**
>
> Thank you for your question. We think the most important part is a good KL-reward tradeoff, and that is why we focus on both KL and reward in this case. According to [1], it is important to gain high reward with a smaller KL, since the gold reward will decrease with large KL. We show that APA is able to achieve better KL-reward tradeoff than PPO in the task of RLHF training.

---

> > ### Comment · Reviewer_GAYy · 2023-11-22
> > **Response after rebuttal**
> >
> > Thanks for your response, which addressed some of my concerns. Now I can see the preliminary results that APA being applied to more general tasks, and the ablation study. I have read other reviewers' comments and updated my score accordingly.
> >
> > I am still with reservation mainly because the small KL between the learned model and the initial model may mean that the language model cannot revise its error anyway. However, I think it is a good start to make the RLHF process more stable.

---

### Meta-Review · Area_Chair_vKE7 · 2023-12-09

**Metareview:**

The paper proposes a new RL technique called advantage induced policy alignment (APA).  It claims that APA is more sample efficient, affords steadier control and has fewer hyperparameters than PPO and AWR.  The paper supports those claims with experiments on finetuning LLMs.  Additional experiments on four simulated robotics were also added.  The claims are also supported with some theory including a finite sample bound that guarantees convergence of the policy at each step to the best regularized policy given the policy of the previous step.

This is very interesting work that is quite promising.  However the reviewers raised several concerns.  I also read the paper and let me elaborate about these concerns.  First, the proposed technique boils down to replacing KL divergence by a Euclidean loss between the target and estimated logits.  At first glance, this does not justify a publication.  In all kinds of algorithms, changing the loss function yields different results, but we don't need a new paper each time someone tries a new loss function.  Unfortunately, the paper does not provide any intuition for why the change in loss function in desirable.  There is empirical and theoretical evidence in favour of the change in loss function, but it would be nice to understand what really makes the proposed loss function better.

Initially, the paper included experiments only on RLHF of LLMs.  A single application is insufficient to claim that the APA is necessarily better. Following the reviewer's recommendation, the paper added simulated robotics experiments in which APA outperformed PPO and AWR in 3/4 domains.  This is good.  However, PPO and AWR are not state of the art algorithms anymore.  When OpenAI did RLHF with PPO, it was understood that PPO was just some RL technique and any RL technique could be used.  So one would expect that APA be compared to state of the art techniques (e.g., SAC).  In addition, the RLHF initially described by OpenAI was fomulated as a combinatorial contextual bandit problem where there is no state transition and therefore this is a simplified class of RL.  The writing of the paper suggests that there are state transitions, but since the state transitions are deterministic and the states are simply the concatenation of the previous state with the next action, this is equivalent to a combinatorial bandit.  Hence, it is awkward to propose a new algorithm and claim that it is better than other algorithms when the main experiments are with respect to the simplified class of combinatorial bandits.  The robotics experiments help to alleviate the concern, but they should be in the main paper.  At the end of the day, the paper needs to choose whether it proposes a new algorithm and claims that is better than previous algorithms with experiments in sebveral applications that compare to state  of the art algorithms, or it focuses on NLP and the claim is about improving RLHF.  For the later an NLP venue might be better, but still one would expect a comparison to state of the art RL/combinatorial bandit techniques.

Theorem 1 is a good step for a theoretical justification of the proposed algorithm.  However, there is a gap between the theorem and the claim that APA is guaranteed to converge to the optimal policy.  The theorem only pertains to one iteration since the optimization is with respect to pi_old.  The theorem shows that with enough roll outs we can guarantee convergence to the best policy that is regularized by pi_old.  But since we change pi_old at each iteration, it is not clear that the overall algorithm will converge to the optimal policy.  This should be discussed and acknowledged.

**Justification For Why Not Higher Score:**

Lack of explanation for why the proposed technique is better, weak experiments and weak theory as explained above.

**Justification For Why Not Lower Score:**

N/A

---

### Decision · Program_Chairs · 2024-01-16

Reject